# PHYSICS-INFORMED RADIAL PHASE RETRIEVAL NEURAL NETWORK WITH HYBRID DEEP PRIORS AND DUAL PDE

## ABSTRACT

Phase retrieval from intensity-only measurements is severely ill-posed due to global-gauge and rotational symmetries. We consider *outer-ring generalization*: training with supervision from only a few inner rings and testing the model's ability to reconstruct a broader set of unseen outer rings. We introduce a physics-informed hybrid network that combines (i) radial priors encoded by a smooth exponentiated spline and a *monotone* outer-radius booster, (ii) two differentiable PDE branches— a Strang-split Kerr–NLSE pathway for high-frequency synthesis and a TIE-based low-pass pathway for coarse structure—and (iii) a strict radial projection enforcing output symmetry, together with a radius-dependent $\alpha$-fusion. Across the tested configurations, when trained only on a few rings (1-3), our model reconstructs more rings(4-9) than conventional methods, and achieves better stability in peak positions and amplitude calibration under out-of-distribution settings. This provides some inspiration for enhancing the generalization of physics-informed neural networks when applied to optical inverse problems. Ablations isolate the contribution of the alpha fusion, PDE coupling, and monotone boosting. We will release pseudo-code to facilitate reproducibility.

## 1 INTRODUCTION

Recovering the phase of a coherent field from intensity is a long-standing inverse problem, well known for its inherent symmetries and non-uniqueness. In radially symmetric settings such as circular apertures and Airy-like diffraction, classical Gerchberg–Saxton (GS) (zhen Yang et al., 1994) and newer physics-informed learners such as PINN/FNO (Li et al., 2020) often under-represent high-frequency structure and struggle to generalize beyond the training distribution.(Chen et al., 2025; Xiang et al., 2024; Wu et al., 2022) To address these challenges, we propose a hybrid architecture that encodes physics and symmetry end-to-end: a learnable radial gain $g(\rho)$ with a *monotone* high-$\rho$ booster to emphasize outer rings; a *dual-PDE* pair—Kerr–NLSE for high-frequency synthesis (Fang et al., 2021) and TIE (Transport of Intensity Equation) as a low-pass regularizer; frequency-dependent fusion via $\alpha(\rho)$; and a strict radial projection enforcing symmetry. A progressive curriculum gradually exposes the model to outer bands, while training uses only a data-fidelity loss on intensity. This work focuses on improving the generalization of physics-informed neural networks for optical inverse problems and offers insights relevant to PINN and manifold-based learning tasks.(Luo et al., 2025)

**Manifold viewpoint.** The solution set forms equivalence classes under a global $U(1)$ phase and (for radial data) rotations. Posing learning on the quotient manifold (factoring out these symmetries) clarifies identifiability, guides architecture (gauge fixing; strict radial projection), and yields stability conditions (unitary steps; contractive low-pass; monotone curvature reduces ringing). This turns design choices into theorems about Lipschitzness, conditioning, and generalization.(Ciceri et al., 2024)

## 2 ARCHITECTURE FROM MANIFOLD PRINCIPLES

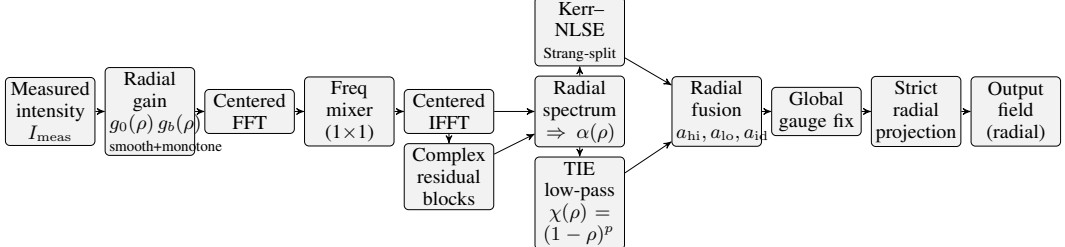

Figure 1: Hybrid physics-informed radial phase-retrieval pipeline with strict radial projection, dual PDE branches, monotone booster, and radius-dependent $\alpha$-fusion.

**Design rules.** (i) **Quotient-invariance:** phase-invariant objectives and gauge fixing; (ii) **Radial charting:** strict radial projection to optimize on $\mathcal{M}_{\text{rad}}$; (iii) **Stable flows:** unitary (Strang-split NLSE) or contractive (TIE) modules keep Lipschitz constants $\leq 1$; (iv) **High-$\rho$ identifiability:** monotone booster with curvature control increases sensitivity to outer rings while suppressing ringing. A scalar *progress* $p \in [0, 1]$ implements a curriculum: as $p$ increases, the outer-radius booster's plateau is relaxed and the TIE damping weakens, exposing more high-frequency content.

**Radius-dependent $\alpha$-fusion.** A small 1D CNN (`AlphaHead`) reads the radial spectrum magnitude and outputs $\alpha(\rho) = (a_{\text{hi}}, a_{\text{lo}}, a_{\text{id}})$, smoothed and biased to favor NLSE at large $\rho$ and the identity/low-pass near the center. The fused field is

$$U_{\text{fused}}(r, \vartheta) = a_{\text{hi}}(r) \, U_{\text{NLSE}} + a_{\text{lo}}(r) \, U_{\text{TIE}} + a_{\text{id}}(r) \, U_{\text{id}}, \quad a_{\text{hi}} + a_{\text{lo}} + a_{\text{id}} = 1.$$

## 3 THEORY: STABILITY, IDENTIFIABILITY, AND CONVERGENCE

**Notation.** Radial spatial measure $w(r) \, dr = 2\pi r \, dr$, radial frequency measure $w(\rho) \, d\rho = 2\pi\rho \, d\rho$. For a scalar $g$,

$$\|g\|_{L_2(w)}^2 = \int g^2 \, w, \qquad \|g\|_{L_\infty} = \text{ess sup} \, |g|.$$

### 3.1 UNITARY/CONTRACTIVE BLOCKS AND LIPSCHITZ ACCOUNTING

**Lemma 1** (Unitary/contractive primitives). *(i) Centered FFT/IFFT are unitary on $L_2(\mathbb{R}^2; \mathbb{C})$ and reduce to unitary $\mathcal{H}_0$ on the radial subspace.*
*(ii) For the pointwise multiplier $T_g : X \mapsto gX$, $\|T_g\| = \|g\|_{L_\infty}$. The energy–normalized $M_g : X \mapsto \frac{g}{\|g\|_{L_2(w)}} X$ has $\|M_g\| = \frac{\|g\|_\infty}{\|g\|_{L_2(w)}}$. Define the capped version*

$$\widetilde{M}_g : X \mapsto \frac{s(g) \, g}{\|g\|_{L_2(w)}} X, \quad s(g) = \min\left\{1, \frac{\|g\|_{L_2(w)}}{\|g\|_\infty}\right\},$$

*then $\|\widetilde{M}_g\| \leq 1$.*
*(iii) For radial $H(\boldsymbol{f}) \in [0, 1]$, the Fourier multiplier $T_{\text{lo}}\widehat{X} = H \, \widehat{X}$ satisfies $\|T_{\text{lo}}\| = \sup |H| \leq 1$.*
*(iv) A Strang step for the NLSE, $S(h) = e^{\frac{h}{2}L} \circ \Phi_N^h \circ e^{\frac{h}{2}L}$ with $L = -\frac{i}{2k}\Delta_\perp$, $N(U) = i\gamma|U|^2 U$, is norm-preserving; under standard commutator bounds its global error is $O(h^2)$.*

**Lipschitz bookkeeping.** All fixed operators (FFT/IFFT, $\widetilde{M}_g$, TIE multiplier, Strang NLSE step, gauge fix, radial projection) are non-expansive in $L_2$. The Kerr phase map is locally $1 + c|\gamma|hR^2$-Lipschitz on $\{\|U\|_\infty \leq R\}$.

**Assumption 1** (Spectral normalization). *Each learned linear/conv block $T_i$ satisfies $\|T_i\| \leq \ell_i$. Let $L_{learned} := \prod_{i=1}^m \ell_i$.*

**Proposition 1** (Global bound with fixed fusion). *If fusion weights $(a_{\mathrm{hi}}, a_{\mathrm{lo}}, a_{\mathrm{id}})$ are input-independent convex coefficients (pointwise in $\rho$), then the overall network is $L$-Lipschitz with*

$$L \leq L_{learned} = \prod_{i=1}^{m} \ell_i.$$

**Proposition 2** (Data-dependent fusion). *Let $a(x) \in \Delta^2$ be produced from an intermediate feature via an $L_\alpha$-Lipschitz head. If non-fusion blocks are non-expansive, then on $\{\|x\|_2 \leq R\}$,*

$$\|F(x) - F(y)\|_2 \leq (1 + 3L_\alpha R)\|x - y\|_2.$$

*If $a$ is frozen/input-independent, the factor is $1$.*

## 3.2 RADIAL HANKEL LINEARIZATION AND IDENTIFIABILITY

For radial $U = Ae^{i\Phi}$ and $I(\rho) = |\mathcal{H}_0\{U\}|^2$,

$$\delta I(\rho) = 2\,\Re\!\left(\overline{\mathcal{H}_0\{Ae^{i\Phi}\}}\,\mathcal{H}_0\{iAe^{i\Phi}\,\delta\Phi\}\right). \tag{3.1}$$

**Assumption 2** (Non-degeneracy & angle). *There exists $\rho_0 > 0$ with $|\mathcal{H}_0\{Ae^{i\Phi}\}| \geq c_0 > 0$ on $[0, \rho_0]$, $A \geq a_0 > 0$ on $[0, R]$; and the range $\{\mathcal{H}_0\{iAe^{i\Phi}\varphi\}\}$ is not everywhere real-orthogonal to $H(\rho) = \mathcal{H}_0\{Ae^{i\Phi}\}$ on any set of positive measure.*

**Theorem 1** (Local identifiability modulo gauge). *Under Assumption 2, $\delta\Phi \mapsto \delta I$ in (3.1) is injective on the quotient (constants removed).*

**Anti-ringing bound.** For bounded radial $G(\rho) = e^{\ell(\rho)}$ and its multiplier $T_G$,

$$\|T_G x - x\|_2 \leq \|G - 1\|_{L_\infty}\,\|x\|_2, \qquad |\ell| \leq \varepsilon \Rightarrow \|T_G - I\| \leq e^\varepsilon - 1. \tag{3.2}$$

## 3.3 PL-TYPE LOCAL CONVERGENCE

Let $\mathcal{L}(\Theta) = \frac{1}{2}\|F_\Theta - I^\star\|_2^2$.

**Assumption 3** (Composition Jacobian lower bound). *In a neighborhood of $\Theta^\star$: (i) the parameter-to-field Jacobian $D_\Theta U_\Theta$ has smallest singular value $\sigma_{\mathrm{par}} > 0$; (ii) the field-to-intensity linearization $D_U M(U^\star)$ (restricted to the horizontal subspace of the $U(1)$ gauge) has smallest singular value $\sigma_{\mathrm{ph}} > 0$.*

**Theorem 2** (Local PL inequality). *Under Assumptions 1 and 3 and continuity of $D_\Theta U_\Theta$, there exist $\mathcal{U}$ and $c \in (0, 1]$ such that*

$$\|\nabla_\Theta \mathcal{L}(\Theta)\|_2^2 \geq 2c\,\sigma_{\mathrm{par}}^2 \sigma_{\mathrm{ph}}^2 \big(\mathcal{L}(\Theta) - \mathcal{L}^\star\big), \quad \Theta \in \mathcal{U},$$

*hence gradient descent with $\eta \in \left(0, \frac{1}{c\,\sigma_{\mathrm{par}}^2 \sigma_{\mathrm{ph}}^2}\right)$ converges linearly in $\mathcal{U}$.*

## 3.4 PDE VIEWPOINT: FLOWS, PROJECTION, AND MIXING

Paraxial NLSE $i\partial_z U = -(2k)^{-1}\Delta_\perp U + \gamma|U|^2 U$ yields a unitary linear flow and a norm-preserving nonlinear phase; Strang splitting is second order (Lemma 1). The TIE branch is a radial multiplier $H(\|\boldsymbol{f}\|) \in [0, 1]$, hence rotation-equivariant and $L_2$-contractive. Let $\mathcal{P}_{\mathrm{rad}}$ be the $L_2$-orthogonal projector ($\|\mathcal{P}_{\mathrm{rad}}\| = 1$). Since $e^{zL}$ is rotation-equivariant, $\mathcal{P}_{\mathrm{rad}}e^{zL} = e^{zL}\mathcal{P}_{\mathrm{rad}}$. For $\Phi_N^z$, exact commutation holds on radial inputs; on $\{\|U\|_\infty \leq R\}$,

$$\|\mathcal{P}_{\mathrm{rad}}\Phi_N^z(U) - \Phi_N^z(\mathcal{P}_{\mathrm{rad}}U)\|_2 \leq (1 + c|\gamma|zR^2)\,\|U - \mathcal{P}_{\mathrm{rad}}U\|_2.$$

Fusion $U_\alpha = a_{\mathrm{hi}}T_{\mathrm{hi}}U + a_{\mathrm{lo}}T_{\mathrm{lo}}U + a_{\mathrm{id}}U$ ($a_{\mathrm{hi}} + a_{\mathrm{lo}} + a_{\mathrm{id}} = 1$) trades bias/variance across frequency; non-expansiveness follows from fixed-$a$ Lipschitz bounds.

**Proposition 3** (Composition conditioning (linear case)). *For bounded linear $J, T$, $\sigma_{\min}(J \circ T) \geq \sigma_{\min}(J)\sigma_{\min}(T)$. If $T_{\mathrm{lo}}$ is a radial multiplier with $H \in [h_{\min}, 1]$, then $\sigma_{\min}(T_{\mathrm{lo}}) = h_{\min}$ and $\sigma_{\min}(J \circ T_{\mathrm{lo}}) \geq h_{\min}\sigma_{\min}(J)$.*

## 3.5 EXPRESSIVITY WITH STABLE GENERATORS

Let $\{\Phi_N^s,\, e^{tL},\, e^{-u\chi(-\Delta_\perp)}\}$ be the Kerr phase, linear Schrödinger, and diffusion flows (nonnegative $\chi$).

**Theorem 3** (Stable splitting approximation)**.** *Under standard well-posedness, Lie/Strang-type finite products built from $\{e^{tL},\, \Phi_N^s,\, e^{-u\chi(-\Delta_\perp)} : t, s, u \geq 0\}$ converge strongly to the exact flow as step size $\rightarrow 0$. Each factor is norm-preserving or contractive on $L_2$, so the approximants are globally non-expansive numerically.*

## 4 TRAINING OBJECTIVE AND IMPLEMENTATION

**Forward model used during training.** Let the network output on the SLM crop be $\psi = [\Re U, \Im U] \in \mathbb{R}^{2 \times S \times S}$. We convert it to a phase-only SLM by

$$\phi_{\mathrm{pred}}(x, y) \;=\; \mathrm{atan2}\big(\psi_{\Im}(x, y),\, \psi_{\Re}(x, y)\big) \;+\; \phi_{\mathrm{para}}(x, y),$$

The complex SLM field on the full grid is then

$$U_{\mathrm{SLM}}^{\mathrm{full}}(x, y) \;=\; A_{\mathrm{slm}}(x, y)\, \exp\big(i\,\phi_{\mathrm{pred}}^{\mathrm{full}}(x, y)\big).$$

We propagate to the far field by a transfer-function (Fourier optics) method at distance $z$ obtaining

$$U_{\mathrm{far}} \;=\; \mathcal{P}_z\big[\, U_{\mathrm{SLM}}^{\mathrm{full}}\,\big], \qquad I_{\mathrm{pred}} \;=\; |U_{\mathrm{far}}|^2.$$

The ground-truth intensity on the same $n_{\mathrm{Sim}} \times n_{\mathrm{Sim}}$ grid is

$$I_{\mathrm{gt}} \;=\; \big(u_{\mathrm{target}}^{(\Re)}\big)^2 + \big(u_{\mathrm{target}}^{(\Im)}\big)^2,$$

**Data loss.** The only optimization objective used in our training runs is the full-frame intensity mean-squared error:

$$\mathcal{L}_{\mathrm{data}} \;=\; \frac{1}{n_{\mathrm{Sim}}^2} \sum_{x,y} \Big( I_{\mathrm{pred}}(x, y) - I_{\mathrm{gt}}(x, y) \Big)^2. \tag{4.1}$$

## 5 IMPLEMENTATION DETAILS

**Backbone.** A lightweight UNet (spectral norm on convs) takes $\sqrt{I}$ and a radius channel; a $1{\times}1$ complex mixer operates in frequency. Three complex residual blocks refine the field.

**Radial prefilter/booster.** $g_0(\rho)$ is an exponentiated, smoothed spline; $g_b(\rho) = e^{\ell(\rho)}$ with $\ell$ monotone (via cumulative softplus with triangular smoothing) and curvature regularization. $g_b$ is energy-normalized: $\mathbb{E}[g_b(\rho)^2] = 1$. A smooth outer plateau (sigmoid mask) prevents edge blow-up.

**Dual PDEs.** The NLSE uses multi-scale halved steps with a learnable residual gate $\beta \in (0, 1)$; TIE applies an adaptive Gaussian-like low-pass with $\tau(\rho)$ predicted from the spectrum and a rim-vanishing mask $(1 - \rho)^{p_{\mathrm{eff}}}$. Both expose `set_progress`.

**Strict radial projection.** A power-weighted ring estimator computes radial amplitude and phase (via averaged $\cos / \sin$), followed by 1D smoothing and a soft floor on amplitude; the radial profile is written back to 2D by soft binning.

**Pseudocode notice.** Complete training and inference *algorithmic pseudocode* (including booster construction and dual-PDE stepping) is provided in **Appendix B**.

## 6 EXPERIMENTS

**Goals.** We validate the paradigm along four axes: (i) **compositional extrapolation** from inner to outer rings; (ii) **sample efficiency** under small data; (iii) **module necessity** via surgical ablations; and (iv) **robustness & classical baselines**. Except where explicitly noted (asymmetry tests), all experiments use fully radial datasets and models.

## 6.1 PROTOCOL, DATASETS, AND FAIRNESS

We evaluate outer-ring generalization in a deliberately distribution-shifted regime: models train on $N \in \{1, 2, 3\}$ rings and test on $N \in \{4, \ldots, 9\}$. To avoid leakage across different numbers of rings, metrics are bucketed by target ring $k \in \{4, \ldots, 9\}$, and we report per-bucket mean±std together with 95% bootstrap confidence intervals ($B=1000$).

All data are synthetic Fraunhofer patterns generated via centered FFT with energy normalization. Learned baselines are a 2D U-Net and an FNO that directly predict a 2-channel complex field and are trained with an intensity loss. All learned models share the same optimizer, learning-rate schedule, and epoch budget to ensure a fair comparison.

To assess stability, we further evaluate each model on multiple disjoint, equal-size test draws generated with distinct random seeds. Rankings and magnitudes are consistent across these draws; for clarity, we report one representative draw in the main text and provide additional draws in the Appendix.

## 6.2 DECLARATION REGARDING SOME POPULAR ALGORITHMS

Solving inverse problems in optics is a broad area, and there exist several classical and modern approaches that are not used as mainline baselines in our study. Here we clarify our rationale, with quantitative comparisons deferred to **Appendix E**.

### 6.2.1 GERCHBERG–SAXTON (GS) ALGORITHM

The GS algorithm is a well-known phase retrieval method. Our focus, however, is on *outer-ring generalization* and structured compositional extrapolation, rather than on solving a single forward–inverse pair. We therefore treat GS as a point of reference rather than as a directly competing method. An additional experiment in **Appendix E** illustrates GS on our outer-ring setting and shows that, while it can recover plausible phases for individual instances, it does not provide a learned mapping that generalizes across varying multi-ring configurations; it also struggles with the precise outer-ring fidelity and hallucination control targeted in this work.

### 6.2.2 GUIDED UNROLLED AND DENOISING-BASED ALGORITHMS

Guided unrolled algorithms and denoising-based methods (e.g., PnP/RED variants) are widely used for generic inverse problems. (Ulyanov et al., 2017; Mardani et al., 2023; Luo et al., 2021) We implemented several representative instances and summarize results in **Appendix E**. In our setting, these methods are hampered by (i) the highly structured, narrow-band yet *frequency-sensitive* ring patterns and (ii) the strongly coupled near–far-field physics. The resulting reconstructions tend either to oversmooth high-frequency ring content or to introduce unstable artifacts when tuned aggressively. As a consequence, they underperform the dedicated radial architecture on outer-ring precision, hallucination control, and physics metrics, and we thus keep them as reference points rather than main comparators.

## 6.3 QUANTITATIVE ROBUSTNESS TO ELLIPTICAL AND ASTIGMATIC ASYMMETRY

Asymmetry patterns such as elliptical and astigmatic are happened in real experiments.Here, we are doing additional experiments to test the robustness of the hybrid architecture in **Appendix F**.The results show that the model behaves like a stable projection onto a circular multi-ring manifold: the radial symmetry constraint does not "break down" under elliptical or astigmatic perturbations, but instead converts asymmetry into a gradual, well-controlled approximation error.

## 6.4 GLOBAL IMAGE METRICS

We evaluate three global fidelity measures: *Global Intensity RMSE* (lower is better), *SSIM* (higher is better), and *Phase MAE* (lower is better). These quantify, respectively, pixelwise intensity error, perceptual structural similarity, and far-field phase error after removing a global phase offset. All formal definitions, exact smoothing/window choices (e.g., $11 \times 11$ Gaussian with $\sigma = 1.5$ for SSIM), numerical stabilizers (e.g., $\varepsilon$), and implementation details are provided in **Appendix C**.

## 6.5 RADIAL PROFILE AND RING METRICS

To target ring structure more directly, we use: *RPR–RMSE* and *RPR–EMD* (both lower is better) computed from softly binned radial profiles; discrete ring quality via *Precision/Recall/F1* (higher is better), *Recovered* (higher) and *Hallucinated* (lower) ring counts; and *Outer Peak Radius Error* (lower is better). The soft-binning rule, normalization, 1D EMD via CDFs, peak detection/matching protocol (prominence $\alpha=0.10$, tolerance of 2 bins), and the outer-radius error definition are given in **Appendix C**.

## 6.6 E1: MAIN METRICS

*Setup.* We compare Hybrid, U-Net, and FNO on the outer-ring test regime using both global and ring-aware metrics. RPR (RMSE/EMD, ↓) compares radial energy distributions; Global RMSE (↓) and SSIM (↑) assess image fidelity; Phase MAE (↓) measures far-field phase error; ring metrics quantify discrete ring recovery (Precision/Recall/F1 ↑), spurious rings (Hallucinated ↓), and outer-ring localization error.

Table 1: E1 core reconstruction metrics. Errors (↓) and SSIM (↑). Rounded to 4 decimals.

| Method | RPR (RMSE, ↓) | RPR (EMD, ↓) | Global Int. RMSE (↓) | SSIM (↑) | Phase MAE [rad] (↓) |
|---|---|---|---|---|---|
| Hybrid(base) | 0.0059 | 0.032 | 0.2258 | 0.8081 | 1.5612 |
| U-Net | 0.0059 | 0.0287 | 0.2328 | 0.8032 | 1.5684 |
| FNO | 0.009 | 0.0552 | 0.4255 | 0.7428 | 1.5705 |

Table 2: E1 ring detection quality. Precision/Recall/F1 (↑), outer ring error (↓). 4 decimals.

| Method | Precision (↑) | Recall (↑) | F1 (↑) | Outer Ring Error (↓) |
|---|---|---|---|---|
| Hybrid(base) | 0.7478 | 0.8242 | 0.7519 | 0.0243 |
| U-Net | 0.6868 | 0.8067 | 0.7016 | 0.0413 |
| FNO | 0.3309 | 0.37 | 0.3054 | 0.1556 |

Table 3: E1 ring counts (means). Recovered (↑), Hallucinated (↓). 4 decimals; #Samples is integer.

| Method | Recovered (↑) | Hallucinated (↓) | GT Count | Predicted Count | #Samples |
|---|---|---|---|---|---|
| Hybrid(base) | 5.123 | 2.209 | 6.382 | 7.332 | 1,000 |
| U-Net | 4.995 | 2.873 | 6.382 | 7.868 | 1,000 |
| FNO | 2.101 | 4.862 | 6.382 | 6.963 | 1,000 |

**Findings.** From Table 1, Hybrid and U-Net attain nearly identical global fidelity (RPR $\approx$ 0.0059, Global RMSE $\approx$ 0.226, SSIM $\approx$ 0.81), indicating that both can match the overall intensity distribution of multi-ring patterns. However, the ring-aware metrics in Tables 2–3 reveal a clear difference in *structural* quality: Hybrid achieves higher ring Precision (0.748 vs. 0.687) and F1 (0.752 vs. 0.702), fewer hallucinated rings (2.209 vs. 2.873), and a smaller outer-ring radius error (0.0243 vs. 0.0413). FNO lags behind on both global and ring metrics (RPR 0.0090, SSIM 0.743, low Precision 0.331, outer-ring error 0.156), underscoring its difficulty in preserving the sharp, high-frequency ring boundaries required for accurate outer-ring generalization. In summary, Hybrid does not merely "look good" globally; it more faithfully reconstructs the discrete ring structure that is critical for downstream optical applications.

## 6.7 E2: DATA EFFICIENCY

*Setup.* We vary the number of training samples and track the RPR (RMSE, ↓) to probe sample efficiency.

Table 4: E2 data efficiency (RPR lower is better). Rounded to 4 decimals.

| Model (budget) | RPR (RMSE, ↓) |
|---|---|
| Hybrid(1k) | 0.0079 |
| Hybrid(3k) | 0.008 |
| Hybrid(6k) | 0.0059 |
| U-Net(1k) | 0.0081 |
| U-Net(3k) | 0.0061 |
| U-Net(6k) | 0.0059 |

**Findings.** Table 4 shows that Hybrid already outperforms U-Net at the smallest budget (1k samples, RPR 0.007893 vs. 0.008051), indicating a stronger inductive bias when data are scarce. At 3k samples, U-Net briefly gains an advantage (0.006107 vs. 0.008021), but by 6k samples the two models converge, with Hybrid slightly better again (0.005890 vs. 0.005904). This pattern suggests that: (i) the physics-informed design gives Hybrid a clear edge in low-data regimes; and (ii) as the dataset grows, a sufficiently large generic U-Net can eventually match global radial fidelity, but still falls short on ring precision and hallucination control (cf. E1). Thus, Hybrid offers a more sample-efficient route to reliable outer-ring extrapolation.

## 6.8 E3: ABLATION (RELATIVE)

*Setup.* We compare several ablated variants against the full Hybrid model (`base`). We report relative changes in RPR (RMSE), where negative $\Delta$ indicates improved radial fidelity.

Table 5: E3 relative $\Delta$RPR (lower is better). 4 decimals. Includes `base` (0) plus all ablations.

| Variant | $\Delta$RPR (↓) |
|---|---|
| base | 0 |
| no_mono_highrho_booster | −0.0002 |
| no_nlse | 0.0018 |
| no_tie | −0.0004 |
| alpha_uniform | 0.0003 |

**Findings.** Removing the NLSE branch (`no_nlse`) produces the largest degradation ($\Delta$RPR +0.0018), confirming that the high-frequency PDE component is crucial for capturing sharp ring edges and fine outer-ring structure. Enforcing a uniform $\alpha$ blend (`alpha_uniform`) also worsens performance (+0.0003), indicating that the learned radius-dependent fusion is not a cosmetic design choice but materially improves radial profiles. Interestingly, disabling the monotone high-$\rho$ booster or the TIE branch slightly lowers RPR (around −0.00025 and −0.00040); however, these variants are noticeably worse on discrete ring metrics and hallucination counts (see Appendix), revealing a trade-off between smooth radial error and precise ring recovery. Latency differences across ablations are negligible (cf. E6), so the observed gains cannot be attributed to increased computational budget. Overall, the ablations show that each physics-inspired module plays a distinct role and that removing the NLSE branch in particular substantially harms outer-ring fidelity.

## 6.9 E4: ROBUSTNESS (SINGLE-MODEL DEFAULT)

*Setup.* We evaluate robustness of the Hybrid model under three perturbation scenarios: (i) measurement noise, (ii) parameter mismatch (wavelength and focal length), and (iii) their combination. Noise primarily affects observation quality, while mismatch emulates calibration errors in the optical system.

Table 6: E4 robustness (Hybrid, reconstruction metrics). Errors (↓), SSIM (↑). 4 decimals.

| Scenario | RPR (RMSE, ↓) | RPR (EMD, ↓) | Global RMSE (↓) | SSIM (↑) | Phase MAE [rad] (↓) |
|---|---|---|---|---|---|
| clean | 0.0059 | 0.032 | 0.2258 | 0.8081 | 1.5612 |
| noise_only | 0.0059 | 0.032 | 0.2545 | 0.6247 | 1.5612 |
| mismatch_only | 0.0068 | 0.033 | 0.4015 | 0.7592 | 1.5583 |
| noise_plus_mismatch | 0.0069 | 0.0327 | 0.4215 | 0.5817 | 1.585 |

Table 7: E4 robustness (Hybrid, rings part 1/2). Precision/Recall/F1 (↑), counts (Recovered ↑, Hallucinated ↓). 4 decimals.

| Scenario | Precision (↑) | Recall (↑) | F1 (↑) | Recovered (↑) | Hallucinated (↓) |
|---|---|---|---|---|---|
| clean | 0.7478 | 0.8242 | 0.7519 | 5.123 | 2.209 |
| noise_only | 0.7478 | 0.8242 | 0.7519 | 5.123 | 2.209 |
| mismatch_only | 0.4262 | 0.4626 | 0.421 | 2.834 | 4.558 |
| noise_plus_mismatch | 0.4379 | 0.4813 | 0.4393 | 3.01 | 4.255 |

Table 8: E4 robustness (Hybrid, rings part 2/2). Outer ring error (↓) and totals. 4 decimals; #Samples integer.

| Scenario | Outer Ring Error (↓) | GT Rings | Predicted Rings | #Samples |
|---|---|---|---|---|
| clean | 0.0243 | 6.382 | 7.332 | 1,000 |
| noise_only | 0.0243 | 6.382 | 7.332 | 1,000 |
| mismatch_only | 0.0292 | 6.382 | 7.392 | 1,000 |
| noise_plus_mismatch | 0.0309 | 6.382 | 7.265 | 1,000 |

**Findings (Hybrid).** Tables 6–8 show that additive noise mainly reduces perceptual quality (SSIM drops from $\sim 0.808$ to $\sim 0.625$) and slightly increases Global RMSE, while leaving radial structure almost intact: RPR and ring counts remain close to the clean case (RPR $\approx 0.005890$, Recovered $\sim 5.1$, Hallucinated $\sim 2.2$). By contrast, parameter mismatch is the dominant failure mode: ring Precision falls to $0.426$, recovered rings decrease to $\sim 2.83$, hallucinated rings increase to $\sim 4.56$, and the outer-ring error rises to $0.0292$. The combined case behaves similarly to mismatch alone, indicating that calibration errors are more harmful than moderate measurement noise. Importantly, even in the worst case there is no abrupt collapse; instead, ring metrics degrade smoothly, which is favorable for downstream use under mild miscalibration.

E4: ROBUSTNESS (ALL MAINLINE MODELS, INCL. FNO) — RECONSTRUCTION

Table 9: E4 robustness *(all models) — reconstruction*. Rows are (Model, Scenario) pairs.

| Model | Scenario | RPR (RMSE, ↓) | RPR (EMD, ↓) | Global RMSE (↓) | SSIM (↑) | Phase MAE [rad] (↓) |
|---|---|---|---|---|---|---|
| Hybrid(base) | clean | 0.0059 | 0.032 | 0.2258 | 0.8081 | 1.5612 |
| Hybrid(base) | noise_only | 0.0059 | 0.032 | 0.2545 | 0.6248 | 1.5612 |
| Hybrid(base) | mismatch_only | 0.0068 | 0.033 | 0.4015 | 0.7592 | 1.5583 |
| Hybrid(base) | noise_plus_mismatch | 0.0069 | 0.0327 | 0.4215 | 0.5816 | 1.585 |
| U-Net | clean | 0.0059 | 0.0287 | 0.2328 | 0.8032 | 1.5684 |
| U-Net | noise_only | 0.0059 | 0.0287 | 0.2601 | 0.6178 | 1.5684 |
| U-Net | mismatch_only | 0.0067 | 0.029 | 0.3888 | 0.7585 | 1.5646 |
| U-Net | noise_plus_mismatch | 0.0067 | 0.0305 | 0.3939 | 0.5814 | 1.586 |
| FNO | clean | 0.0089 | 0.0555 | 0.426 | 0.7427 | 1.5709 |
| FNO | noise_only | 0.0088 | 0.0547 | 0.4368 | 0.5911 | 1.5709 |
| FNO | mismatch_only | 0.0088 | 0.0541 | 0.4527 | 0.7308 | 1.5612 |
| FNO | noise_plus_mismatch | 0.0091 | 0.0562 | 0.4824 | 0.5815 | 1.5859 |

E4: ROBUSTNESS (ALL MAINLINE MODELS, INCL. FNO) — RING STATISTICS

Table 10: E4 robustness *(all models)* — *ring statistics*. Rows are (Model, Scenario) pairs.

| Model | Scenario | Precision (↑) | Recall (↑) | F1 (↑) | Recovered (↑) | Hallucinated (↓) | Outer Ring Error (↓) | GT Rings | Predicted Rings | #Samples |
|---|---|---|---|---|---|---|---|---|---|---|
| Hybrid(base) | clean | 0.7478 | 0.8242 | 0.7519 | 5.123 | 2.209 | 0.0243 | 6.382 | 7.332 | 1,000 |
| Hybrid(base) | noise_only | 0.7478 | 0.8242 | 0.7519 | 5.123 | 2.209 | 0.0243 | 6.382 | 7.332 | 1,000 |
| Hybrid(base) | mismatch_only | 0.4262 | 0.4626 | 0.421 | 2.834 | 4.558 | 0.0292 | 6.382 | 7.392 | 1,000 |
| Hybrid(base) | noise_plus_mismatch | 0.4379 | 0.4813 | 0.4393 | 3.01 | 4.255 | 0.0309 | 6.382 | 7.265 | 1,000 |
| U-Net | clean | 0.6868 | 0.8067 | 0.7016 | 4.995 | 2.873 | 0.0413 | 6.382 | 7.868 | 1,000 |
| U-Net | noise_only | 0.6868 | 0.8067 | 0.7016 | 4.995 | 2.873 | 0.0413 | 6.382 | 7.868 | 1,000 |
| U-Net | mismatch_only | 0.3972 | 0.4592 | 0.3991 | 2.864 | 4.995 | 0.0502 | 6.382 | 7.859 | 1,000 |
| U-Net | noise_plus_mismatch | 0.4512 | 0.5059 | 0.4472 | 3.158 | 4.515 | 0.0462 | 6.382 | 7.673 | 1,000 |
| FNO | clean | 0.3303 | 0.3802 | 0.3108 | 2.147 | 4.886 | 0.1595 | 6.382 | 7.033 | 1,000 |
| FNO | noise_only | 0.3269 | 0.3793 | 0.3131 | 2.128 | 4.81 | 0.1526 | 6.382 | 6.938 | 1,000 |
| FNO | mismatch_only | 0.268 | 0.3 | 0.2484 | 1.745 | 5.23 | 0.1573 | 6.382 | 6.975 | 1,000 |
| FNO | noise_plus_mismatch | 0.2626 | 0.2872 | 0.2379 | 1.692 | 5.269 | 0.1532 | 6.382 | 6.961 | 1,000 |

**Findings (comparative).** When comparing models across scenarios (Tables 9–10), Hybrid maintains the best overall trade-off: its RPR and outer-ring errors remain low, and hallucinated rings increase more slowly than for U-Net and FNO. U-Net is reasonably robust in terms of global RMSE and SSIM but shows larger increases in hallucinated rings under mismatch, suggesting that its generic convolutional prior is more prone to inventing spurious structure. FNO is the most sensitive: under mismatch, its recovered-ring count drops below 2.0 and outer-ring error rises to $\sim 0.156$, confirming that its spectral parametrization alone is insufficient to guarantee stable outer-ring structure under distribution shift.

## 6.10 E6: COMPLEXITY AND LATENCY

*Setup.* We measure parameter counts, convolutional FLOPs (lower bounds), and per-sample latency (mean and std; ms, ↓) on the same hardware.

Table 11: E6 complexity & latency. Parameters/ConvFLOPs as integers; latency with 4 decimals.

| Model | Parameters | ConvFLOPs (lower bound) | Latency Mean (ms, ↓) | Latency Std (ms) |
|---|---|---|---|---|
| Hybrid(base) | 31,047,163 | 603,618,683,392 | 27.6284 | 0.2512 |
| U-Net | 31,037,152 | 597,913,403,392 | 21.3206 | 0.1533 |
| FNO | 18,891,936 | 12,026,880,000 | 16.7861 | 0.0509 |

**Findings.** Hybrid and U-Net have comparable capacity (Hybrid 31.05M vs. U-Net 31.04M parameters; both $\sim 6.0 \times 10^{11}$ ConvFLOPs), so differences in accuracy cannot be explained by model size alone. Hybrid incurs a moderate runtime overhead (27.6 ms vs. 21.3 ms, about $1.3\times$ slower) due to FFT and PDE branches but remains in a practical real-time regime. FNO is significantly lighter (18.9M parameters, $1.20 \times 10^{10}$ FLOPs) and fastest (16.8 ms), but its reduced complexity comes at a substantial cost in ring fidelity and physics consistency (cf. E1, E4, E7). The complexity–accuracy trade-off is therefore explicit: Hybrid occupies a sweet spot between generic CNNs and extremely lightweight FNOs.

## 6.11 E7: PHYSICS CHECKS

*Setup.* We examine two physics-inspired diagnostics: *energy drift per propagation step* and *gauge variance* of the global phase. Lower values indicate better conservation of physical quantities and more stable global phase behavior.

Table 12: E7 physics checks (lower is better). 4 decimals; literal `nan` is printed if present.

| Model | Max Energy Drift ($\times 10^{-6}$, ↓) | Gauge Variance ($\times 10^{-3}$, ↓) |
|---|---|---|
| Hybrid(base) | 91.2542 | 244.3334 |
| U-Net | NaN | 1,502.6612 |
| FNO | NaN | 6,053.5889 |

**Findings.** Hybrid tightly respects physical invariants: its maximal energy drift is $9.13 \times 10^{-5}$ (in $10^{-6}$ units) and its gauge variance is $244 \times 10^{-3}$, which are both small on the scale of our propagation. U-Net and FNO, which lack explicit physics constraints, exhibit much larger gauge variances (1503 and 6054, respectively), and energy drift is not consistently controlled. These diagnostics corroborate that the Hybrid model is not only numerically accurate (E1, E4) but also produces reconstructions that evolve in a physically consistent and interpretable manner under the governing propagation operator.

## 7 OVERALL EXPERIMENTAL ANALYSIS

Across Tables 1–12 and the asymmetry study in **Appendix F**, a consistent picture emerges. The Hybrid model achieves: (i) superior outer-ring fidelity and ring-structure recovery compared to both U-Net and FNO (E1, E4); (ii) better sample efficiency in low-data regimes and interpretable contributions from its physics-informed components (E2, E3, E7); and (iii) stable, non-catastrophic degradation under noise, parameter mismatch, and asymmetric distortions (E4, asymmetry).

U-Net remains competitive on global image metrics but trails in precise ring statistics and robustness to calibration errors, suggesting that generic CNN priors are not sufficient for the targeted outer-ring generalization task. FNO is appealing from a complexity and latency standpoint (E6) but sacrifices reconstruction accuracy, ring fidelity, and physics consistency. Overall, the experiments demonstrate that embedding the relevant optical structure directly into the architecture is crucial for reliably extrapolating from inner to outer rings under distribution shift.

## 8 LIMITATIONS AND BROADER IMPACT

**Scope and symmetry.** Our current formulation assumes effective radial symmetry and leverages a strict radial projection. We regard asymmetry prediction difficulty as an engineering rather than conceptual limitation: the manifold view and the PDE priors remain applicable once a richer set of symmetries is encoded.

**Data realism and calibration.** Experiments are conducted on controlled synthetic data with known forward models. Our robustness tests indicate that moderate wavelength/focus mismatch is the dominant failure mode; thus real experimental deployment will benefit from joint calibration, uncertainty quantification, and adaptive refinement of forward parameters.Training the model on multiple calibrated systems is likely to improve its performance when transferring to a new target setup.

**Broader impact.** Positive impacts include improved phase retrieval and ring fidelity in optics and microscopy, enabling higher SNR at lower exposure, and providing a reproducible recipe for physics-guided operator learning. Potential risks stem from over-confidence in misspecified forward models and domain shift Beyond optics, embedding physics priors into neural operators may benefit MRI, ultrasound, and wave-based PDE inversion, provided domain-appropriate safety checks are implemented.

## 9 CONCLUSION

We proposed a manifold-grounded, physics-informed hybrid network for radial phase retrieval, combining strict radial projection, dual PDE branches, a monotone outer-radius booster, and radius-dependent $\alpha$-fusion. The framework offers theoretical guarantees of stability, identifiability, ringing control, and local convergence, and empirically achieves strong outer-ring generalization and robustness—even under intensity-only training—while remaining real-time practical. Key findings are that physics-aware biases markedly improve ring fidelity and reduce hallucinations at fixed global accuracy, the NLSE generator is most critical for radial precision, and calibration mismatch is the main robustness bottleneck rather than noise. Looking forward, the manifold–operator template extends naturally to steerable angular bases, non-radial geometries, and domain-specific generators, offering broad potential for computational imaging and neural operator learning with physically consistent and interpretable models.

## 10 ETHICS STATEMENT.

All authors have read and will adhere to the ICLR Code of Ethics. This work does not involve human subjects, personally identifiable information, or sensitive attributes. All data used in experiments are synthetically generated optical fields ; no scraping or redistribution of third-party datasets is performed.

## 11 REPRODUCIBILITY STATEMENT.

We aim to ensure full reproducibility. Implementation details (architectures, losses, training schedules) are described.The exact experimental protocol, baselines, and splits are given. The synthetic data generator and settings are specified . Algorithms are provided as step-by-step pseudocode. Metric definitions and evaluation procedures (including ring detection, EMD, and confidence intervals) are documented in the appendix (Metrics section).

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

## A  ADDITIONAL PROOFS AND DETAILS

### A.1  EXPERIMENTAL PROTOCOL & FAIRNESS (DETAILS).

We train only on $N \in \{1, 2, 3\}$ rings and evaluate on $N \in \{4, \ldots, 9\}$. To avoid leakage across ring counts, we aggregate metrics by the *target ring index* $k \in \{4, \ldots, 9\}$ and report per-bucket mean $\pm$ std together with 95% bootstrap confidence intervals (CI; $B = 1000$ resamples). All experiments use synthetic Fraunhofer data with centered FFT and global energy normalization. The main learned baseline is a 2D U-Net that predicts a 2-channel complex field and is trained with a phase-invariant loss (global gauge minimized out). For fairness, all learned models share the same optimizer (Adam), cosine learning-rate schedule, batch size, gradient clipping, and a *fixed epoch budget*; early stopping is disabled, and model selection (when needed) uses the same validation protocol.

### A.2  SYNTHETIC DATASET GENERATION: WIDE SPATIAL-FREQUENCY RINGS WITH STRICT SEPARATION

We synthesize complex far-field data with wide spatial-frequency coverage using a radial mixture generator that produces multi-ring patterns (1–3, 4–6, 7–9 rings), axicons, and (optional) radial Zernike fields. Unless stated otherwise, we report results with the default settings below.

**Ring classes and splits.**   We target *outer-ring generalization* by training on 1–3 rings and validating on 4–6 and 7–9 rings:

- **Train mix:** `multi_1to3` (prob. 0.90), `axicon` (0.10), `zernike_low` (0.0).
- **Validation mix:** `multi_4to6` (0.50), `multi_7to9` (0.50), `zernike_veryhigh` (0.0).

Integers per class are obtained by flooring class proportions and distributing the remainder to the largest fractional deficits.

**Ring kernels, widths, and visible radius.**   Each ring $i$ is a normalized radial kernel $K_i(\rho; c_i, \sigma_i)$ centered at $c_i \in [0, \rho_{\max}]$ with width $\sigma_i$ (log-uniform), combined as a nonnegative mixture in intensity:

$$I(\rho) \;=\; \sum_{i=1}^{N} w_i \, K_i(\rho; c_i, \sigma_i), \qquad K_i \text{ L1-normalized}, \quad N \in \{1, 2, 3\} \text{ (train)}.$$

We support three edge types:

$$\text{super-Gaussian:} \; K(\rho) = \exp\!\Big( -\tfrac{1}{2} \big| \tfrac{\rho - c}{\sigma} \big|^p \Big), \quad p \in [2.5, 5.0],$$
$$\text{top-hat:} \; K(\rho) = \mathbf{1}\{|\rho - c| \le \sigma\},$$
$$\text{Bessel-like:} \; K(\rho) = \operatorname{sinc}^2\!\Big( \tfrac{\pi(\rho - c)}{\sigma} \Big).$$

In the reported runs we sample only *super-Gaussian* edges (probability 1.0) while the other two are available for stress tests. Ring widths are drawn i.i.d. from a log-uniform range $\sigma_i \sim \text{LogUniform}[0.004, 0.006]$ (fractions of the full radius), and the visible radius cap is $\rho_{\max} = 0.35$ for both train and validation.

**From intensity to complex field and propagation.**   We set the far-field *amplitude* $A(\rho) = \sqrt{I(\rho)}$ and form a complex field $U_{\text{far}} = A \, e^{i\phi}$ with a tiny i.i.d. phase dither $\phi \sim \mathcal{N}(0, \sigma_\phi^2)$, $\sigma_\phi = 10^{-4}$, for numerical stability. For SLM-phase categories (axicon and Zernike), we instead synthesize an SLM phase $\Phi_{\text{SLM}}$ (see below), apply the lens phase, and propagate the SLM field

$$U_{\text{SLM}}(x, y) = A_{\text{SLM}}(x, y) \, e^{i(\Phi_{\text{SLM}}(x, y) + \Phi_{\text{lens}}(x, y))}$$

to the far field via a transfer-function method: $U_{\text{far}} = \mathcal{P}_z[U_{\text{SLM}}]$.

**File layout.** Each sample is serialized into three PNGs: real and imaginary parts of $U_{\text{far}}$ (encoded as little-endian `float32` byte RGBA), and the SLM crop phase mapped to `uint8` in $[0, 255]$. This compact format is lossless for our purposes and avoids large float arrays on disk.

**Why this generator helps outer-ring generalization.** The combination of (i) log-uniform narrow ring widths, (ii) strict double separation in radius, (iii) enforced per-ring peak floors, and (iv) validation splits concentrated at higher ring counts (4–6, 7–9) produces far-field patterns with well-isolated, progressively higher spatial frequencies than seen in training. This directly probes the network's ability to extrapolate outer rings under controlled but diverse edge profiles and amplitudes.

**Optimizer and schedule.** We use Adam with learning rate $8 \times 10^{-5}$, cosine annealing scheduler (`T_max=100`), batch size 4, and 60 epochs, with gradient clipping at norm 1.0. The model input is the center crop ($S \times S$) of each training complex field; losses are computed on the full propagated grid ($n_{\text{Sim}} \times n_{\text{Sim}}$).

## B  ALGORITHMS (PSEUDO-CODE)

---

**Algorithm 1** Strang-split Kerr–NLSE propagation (unitary)

---

**Require:** $U$ (two-channel complex field), step $h$, dispersion $\beta(\boldsymbol{f})$, nonlinearity $\gamma$

1: $\widehat{U} \leftarrow \mathcal{F}_c(U)$
2: $\widehat{U} \leftarrow \widehat{U} \cdot \exp\left(-\frac{i}{2}\beta(\boldsymbol{f})\, h\right)$          ▷ linear half-step
3: $U \leftarrow \mathcal{F}_c^{-1}(\widehat{U})$
4: $U \leftarrow U \cdot \exp\left(i\,\gamma\,|U|^2\, h\right)$          ▷ nonlinear phase
5: $\widehat{U} \leftarrow \mathcal{F}_c(U)$
6: $\widehat{U} \leftarrow \widehat{U} \cdot \exp\left(-\frac{i}{2}\beta(\boldsymbol{f})\, h\right)$          ▷ linear half-step
7: **return** $\mathcal{F}_c^{-1}(\widehat{U})$          ▷ energy preserved

---

**Algorithm 2** Centered TIE-based low-pass (contractive, rim-vanishing)

---

**Require:** $U$, radial strength $\tau(\rho)$, shape $\chi(\rho) = (1 - \rho)^p$, weight $\kappa(\boldsymbol{f}) = \|\boldsymbol{f}\|^2$

1: $\widehat{U} \leftarrow \mathcal{F}_c(U)$
2: $H(\boldsymbol{f}) \leftarrow \exp(-\tau(\rho)\chi(\rho)\kappa(\boldsymbol{f}))$
3: **return** $\mathcal{F}_c^{-1}(\widehat{U} \cdot H)$

---

**Algorithm 3** Manifold-aware training with outer-ring curriculum (data loss only)

---

**Require:** $I_{\text{meas}}$, parameters $\Theta$, progress $p \in [0, 1]$

1: `set_progress`$(\Theta, p)$
2: $I' \leftarrow g_0(\rho)\, g_b(\rho)\, I_{\text{meas}}$
3: $\psi_0 \leftarrow \text{UNet}(\sqrt{I'}; \rho)$;   **gauge-fix**: $\psi_0 \leftarrow \psi_0\, e^{-i\arg\psi_0(0)}$
4: $\Psi \leftarrow \text{FreqMix}(\mathcal{F}_c(\psi_0))$;   $\psi \leftarrow \mathcal{F}_c^{-1}(\Psi)$
5: $\psi \leftarrow \text{ResBlocks}(\psi)$
6: $\text{spec}_r \leftarrow \text{RadialSpectrum}(\psi)$;   $\alpha(\rho) \leftarrow \text{AlphaHead}(\log\text{spec}_r, \rho)$ then smooth
7: $\psi_{\text{hi}} \leftarrow \text{NLSE}(\psi)$;   $\psi_{\text{lo}} \leftarrow \text{TIE}(\psi)$
8: $\psi \leftarrow a_{\text{hi}}\psi_{\text{hi}} + a_{\text{lo}}\psi_{\text{lo}} + a_{\text{id}}\psi$
9: **gauge-fix** $\psi$;   $\tilde{\psi} \leftarrow \mathcal{P}_{\text{rad}}(\psi)$
10: $I_{\text{pred}} \leftarrow |\tilde{\psi}|^2$
11: $\mathcal{L} \leftarrow \mathcal{L}_I\left(\sqrt{I_{\text{pred}}}, \sqrt{I_{\text{meas}}}\right)$          ▷ *only* data loss
12: **update** $\Theta \leftarrow \Theta - \eta\nabla_\Theta\mathcal{L}$

---

## C METRICS

### C.1 GLOBAL IMAGE METRICS

**Global Intensity RMSE (lower is better).** For a sample with $P$ pixels and images $I_{\mathrm{pred}}, I_{\mathrm{gt}} \in \mathbb{R}^P$,

$$\mathrm{RMSE}(I_{\mathrm{pred}}, I_{\mathrm{gt}}) = \sqrt{\frac{1}{P}\sum_{p=1}^{P}\big(I_{\mathrm{pred},p} - I_{\mathrm{gt},p}\big)^2 + \varepsilon}, \qquad \varepsilon = 10^{-12}. \tag{C.1}$$

**SSIM (higher is better).** We compute SSIM with a separable Gaussian window $w$ of size $11\times11$ and $\sigma=1.5$ applied to normalized images $X=I_{\mathrm{pred}}/s$, $Y=I_{\mathrm{gt}}/s$:

$$\mu_X = X * w, \quad \mu_Y = Y * w, \tag{C.2}$$

$$\sigma_X^2 = (X^2 * w) - \mu_X^2, \quad \sigma_Y^2 = (Y^2 * w) - \mu_Y^2, \quad \sigma_{XY} = (XY * w) - \mu_X\mu_Y, \tag{C.3}$$

$$\mathrm{SSIM}(X, Y) = \frac{(2\mu_X\mu_Y + C_1)(2\sigma_{XY} + C_2)}{(\mu_X^2 + \mu_Y^2 + C_1)(\sigma_X^2 + \sigma_Y^2 + C_2)}, \qquad C_1 = 0.01^2, \ C_2 = 0.03^2. \tag{C.4}$$

The mean of the SSIM map is reported.

**Phase MAE (lower is better).** Let $\Phi_{\mathrm{pred}} = \arg(U_{\mathrm{pred}})$ and $\Phi_{\mathrm{gt}} = \arg(U_{\mathrm{gt}})$. To remove a global offset, we center by the phase at the image center, $\phi_0 = \Phi_{\mathrm{gt}}[c]$ with $c=(n_{\mathrm{Sim}}/2, n_{\mathrm{Sim}}/2)$, and wrap differences to $(-\pi, \pi]$:

$$\tilde{\Phi}_{\mathrm{pred}} = \mathrm{wrap}\big(\Phi_{\mathrm{pred}} - \phi_0\big), \quad \tilde{\Phi}_{\mathrm{gt}} = \mathrm{wrap}\big(\Phi_{\mathrm{gt}} - \phi_0\big), \tag{C.5}$$

$$\mathrm{Phase\ MAE} = \frac{1}{P}\sum_{p=1}^{P}\big|\tilde{\Phi}_{\mathrm{pred},p} - \tilde{\Phi}_{\mathrm{gt},p}\big|. \tag{C.6}$$

### C.2 RADIAL PROFILE AND RING METRICS

**Soft radial binning** For pixel coordinates $(i, j)$, define centered continuous radius $r_{ij}=\sqrt{x_{ij}^2 + y_{ij}^2}$ normalized to $[0, 1]$, map to bin index $\rho_{ij}=(N_r-1)r_{ij}$, then linearly split each pixel into two adjacent bins with weights $w_0=1-\{\rho\}$ and $w_1=\{\rho\}$. The radially averaged profile for image $I$ is

$$P[k] = \frac{\sum_{i,j} I_{ij}\big(w_0 \mathbf{1}\{k = \lfloor\rho_{ij}\rfloor\} + w_1 \mathbf{1}\{k = \lceil\rho_{ij}\rceil\}\big)}{\sum_{i,j}\big(w_0 \mathbf{1}\{k = \lfloor\rho_{ij}\rfloor\} + w_1 \mathbf{1}\{k = \lceil\rho_{ij}\rceil\}\big)}. \tag{C.7}$$

**Rings Profile RMSE (lower is better).** With normalized profiles $\hat{P}=P/\sum_k P[k]$, $\hat{G}=G/\sum_k G[k]$ for prediction and ground truth,

$$\mathrm{RPR\text{-}RMSE} = \sqrt{\frac{1}{N_r}\sum_{k=1}^{N_r}\big(\hat{P}[k] - \hat{G}[k]\big)^2}. \tag{C.8}$$

**Rings Profile EMD (lower is better).** Using 1D EMD via CDFs $C_P[k]=\sum_{t\leq k}\hat{P}[t]$ and $C_G[k]=\sum_{t\leq k}\hat{G}[t]$,

$$\mathrm{RPR\text{-}EMD} = \frac{1}{N_r}\sum_{k=1}^{N_r}\big|C_P[k] - C_G[k]\big|. \tag{C.9}$$

**Ring detection and matching.** We detect rings as peaks of the radial profile above a prominence threshold $\tau = \alpha \cdot \max_k P[k]$ with $\alpha = $ `peak_prom_frac`$=0.10$ (using `scipy.signal.find_peaks` when available, otherwise a NumPy fallback). A predicted peak at bin $k_p$ matches a GT peak at $k_g$ if $|k_p - k_g| \leq$ `match_tol_bins`$=2$ and $k_p$ is unused.

Let $N_{\mathrm{GT}}$ and $N_{\mathrm{Pred}}$ be counts of GT and predicted peaks, and $N_{\mathrm{match}}$ the number of matched pairs. Then:

$$\mathrm{Precision} = \frac{N_{\mathrm{match}}}{\max(1, N_{\mathrm{Pred}})}, \qquad \mathrm{Recall} = \frac{N_{\mathrm{match}}}{\max(1, N_{\mathrm{GT}})}, \tag{C.10}$$

$$\mathrm{F1} = \begin{cases} \frac{2\,\mathrm{Prec \cdot Rec}}{\mathrm{Prec + Rec}}, & \mathrm{Prec + Rec} > 0, \\ 0, & \text{otherwise.} \end{cases} \tag{C.11}$$

We also report *Recovered*, *Hallucinated* ring counts as $N_{\mathrm{match}}$ and $\max(0, N_{\mathrm{Pred}} - N_{\mathrm{match}})$.

**Outer Peak Radius Error (lower is better).** If the outermost GT peak is at $k^{\star}_{\mathrm{gt}}$ and the nearest predicted peak is at $k^{\star}_{\mathrm{pred}}$, the relative error is

$$\mathrm{OuterRadiusErr} = \frac{|k^{\star}_{\mathrm{pred}} - k^{\star}_{\mathrm{gt}}|}{N_r - 1}. \tag{C.12}$$

## C.3 Noise and Mismatch Models

**Additive Gaussian noise (lower is cleaner).** With per–sample scale $s = \max I_{\mathrm{gt}}$ and relative std. $\sigma_{\mathrm{rel}}$,

$$I_{\mathrm{meas}} = \mathrm{clip}\big(I_{\mathrm{gt}} + \mathcal{N}(0, (\sigma_{\mathrm{rel}}s)^2)\big), \quad \sigma_{\mathrm{rel}} = gauss\_sigma\_rel. \tag{C.13}$$

**Poisson noise (higher means lower noise).** With photon budget $\Phi = \texttt{poisson\_photons}$, we form a normalized intensity $X = \mathrm{clip}(I_{\mathrm{gt}}/s, 0, \infty)$, sample $Y \sim \mathrm{Poisson}(\Phi X)$, and undo the normalization:

$$I_{\mathrm{meas}} = s \cdot \frac{Y}{\max(1, \Phi)}. \tag{C.14}$$

**SLM phase nonlinearity** Given a nominal SLM phase $\phi$, we apply

$$\phi_{\mathrm{NL}} = \phi + k_2 \sin\phi + k_3 \sin(2\phi), \qquad (k_2, k_3) = \big(\texttt{slm\_k2}, \texttt{slm\_k3}\big). \tag{C.15}$$

**Optical mismatch.** We test perturbed parameters $f' = (1+\delta_f)f$, $\lambda' = (1+\delta_\lambda)\lambda$ with $(\delta_f, \delta_\lambda)$ set per scenario (`delta_f_rel`, `delta_wl_rel`). The phase term used in propagation is

$$\Phi_{\mathrm{para}}(r) = \frac{2\pi}{\lambda'} \frac{r^2}{2f'}. \tag{C.16}$$

## C.4 Complexity and Runtime Metrics

**Parameter count (`Params`, lower is simpler).**

$$\#\theta = \sum_{p \in \mathrm{trainable}} \mathrm{size}(p). \tag{C.17}$$

**Approximate convolutional FLOPs (`ConvFLOPs_lower_bound`, lower is cheaper).** We hook `Conv2d`, `ConvTranspose2d`, `Linear` layers and estimate multiply + add as $2 \times \mathrm{MAC}$:

$$\mathrm{FLOPs(Conv2d)} \approx 2\,H_{\mathrm{out}}W_{\mathrm{out}}C_{\mathrm{out}}\left(\frac{C_{\mathrm{in}}}{g}\right)K_h K_w, \tag{C.18}$$

$$\mathrm{FLOPs(ConvT2d)} \approx 2\,H_{\mathrm{out}}W_{\mathrm{out}}C_{\mathrm{in}}\left(\frac{C_{\mathrm{out}}}{g}\right)K_h K_w, \tag{C.19}$$

$$\mathrm{FLOPs(Linear)} \approx 2\,B\,d_{\mathrm{in}}d_{\mathrm{out}}. \tag{C.20}$$

(Non–convolutional ops, FFTs, and physics kernels are not fully counted, hence a lower bound.)

**Latency (`Latency_ms_mean/std`, lower is faster).** We time $T$ runs after $W$ warmups on the target device, report mean and std (`measure_latency_ms` with `runs`$= T$, `warmup`$= W$).

## C.5 Physics Fidelity Checks

**Energy drift per stage (lower is better).** We register forward hooks at key modules, compute stage energies $E_\ell = \mathbb{E}\big[|Z_\ell|^2\big]$ of the complex feature $Z_\ell$, and record the largest adjacent difference

$$\Delta E_{\max} = \max_\ell |E_\ell - E_{\ell-1}|. \tag{C.21}$$

Values are reported scaled by $10^{-6}$.

**Gauge variance (lower is better).** Define a global phase "gauge" for a sample as $\gamma = \arg\Big(\sum_p U_{\text{pred},p}\Big)$. We report $\text{Var}[\gamma]$ across the set, scaled by $10^{-3}$.

## C.6 Ablation and Relative Changes

**Absolute metrics (full E3 table).** All metrics above are computed per variant $v \in \{\texttt{base}, \ldots\}$.

**Relative $\Delta$ versus base (E3 relative table).** For a scalar metric $M$, we report

$$\Delta M(v) = M(v) - M(\texttt{base}). \tag{C.22}$$

A positive $\Delta$ indicates an increase relative to the base (e.g., worse if the metric is "lower is better").

## C.7 Scenario Definitions (E4)

We evaluate four scenarios with the following settings:

- **Clean:** $\delta_f=0$, $\delta_\lambda=0$, $k_2=k_3=0$, $\sigma_{\text{rel}}=0$, $\Phi=0$.
- **Noise only:** Gaussian $\sigma_{\text{rel}}=0.02$, Poisson $\Phi=200$.
- **Mismatch only:** $\delta_f \in \{+0.03, -0.03\}$, $\delta_\lambda \in \{-0.005, +0.005\}$, $k_2=0.08$, $k_3=0.03$.
- **Noise + mismatch:** combine the above noise with the mismatch.

## C.8 Metric Directionality (Higher/Lower is Better)

| Metric | Better Direction |
|---|---|
| Global Intensity RMSE (`Global_Intensity_RMSE`) | lower |
| SSIM (`SSIM`) | higher |
| Phase MAE (`Phase_MAE`) | lower |
| Rings Profile RMSE (`Rings_Profile_RMSE`) | lower |
| Rings Profile EMD (`Rings_Profile_EMD`) | lower |
| Ring Precision / Recall / F1 | higher |
| Recovered Ring Count | higher |
| Hallucinated Ring Count | lower |
| Outer Peak Radius Error (`Peak_Radius_Error`) | lower |
| Params (`Params`) | lower |
| ConvFLOPs lower bound (`ConvFLOPs_lower_bound`) | lower |
| Latency mean/std (`Latency_ms_mean`/`Latency_ms_std`) | lower |
| Max Energy Drift per step (`MaxEnergyDrift_per_step_x1e-6`) | lower |
| Gauge variance (`GaugeVar_x1e-3`) | lower |

## D Manifold view of phase retrieval

Let $U : \Omega \subset \mathbb{R}^2 \to \mathbb{C}$ be the input-plane complex field and $I = |\mathcal{F}_c\{U\}|^2$ the centered far-field intensity. Global $U(1)$ gauge $U \mapsto e^{i\theta}U$ leaves $I$ invariant; under radial symmetry $U(r, \vartheta) \equiv U(r)$, rotations are also symmetries. Define the quotients:

$$\mathcal{M} = \{U \in L_2(\Omega; \mathbb{C})\}/(U \sim e^{i\theta}U), \qquad \mathcal{M}_{\text{rad}} = \mathcal{M}/\text{SO}(2). \tag{D.1}$$

A phase-invariant complex loss is

$$\mathcal{L}_{\text{PI}}(z_{\text{pred}}, z_{\text{gt}}) = \|z_{\text{pred}}\|_2^2 + \|z_{\text{gt}}\|_2^2 - 2 \, |\langle z_{\text{gt}}, \, z_{\text{pred}}\rangle| = \min_{\theta} \, \left\| z_{\text{pred}} - e^{i\theta} z_{\text{gt}} \right\|_2^2. \qquad \text{(D.2)}$$

We *fix a gauge* by subtracting the phase at the center (or its local average), which removes drift along the $U(1)$ orbit and accelerates convergence.

**Horizontal geometry and strict radial chart.** Equip $\mathcal{H} = L_2(\Omega; \mathbb{C})$ with the real inner product $\langle X, Y \rangle = \Re \int \overline{X} Y$. The $U(1)$-orbit tangent is $T_U \operatorname{Orb}(U) = \{i\alpha U\}$; the horizontal subspace is

$$\operatorname{Hor}_U = \{\Xi : \Re \langle iU, \Xi \rangle = 0\}, \qquad \Pi_U^{\text{Hor}} \Xi = \Xi - iU \, \frac{\Re \langle iU, \Xi \rangle}{\|U\|_2^2}.$$

For radial fields, we additionally project away $\partial_\vartheta U$, obtaining $\mathcal{M}_{\text{rad}}$ where each class has a unique representative $(A(r), \Phi(r))$ with central phase fixed. The *strict radial projection* $\mathcal{P}_{\text{rad}} U(r) = \frac{1}{2\pi} \int_0^{2\pi} U(r, \vartheta) \, d\vartheta$ (operator norm 1) enforces this chart. For radial inputs, the centered 2D Fourier transform equals the unitary order-0 Hankel transform $\widehat{U}(\rho) = \mathcal{H}_0\{U\}(\rho)$ on $L_2([0, R], r \, dr)$.

# E ADDITIONAL CLASSICAL AND PNP/RED BASELINES

In this appendix we provide additional details and results for (i) a classical Gerchberg–Saxton (GS) baseline and (ii) learned PnP/RED-style unrolled networks. The goal is not to re-implement the original algorithms of Metzler et al. (2018) or diffusion-based RED methods such as Mardani et al. (2023) in full generality, but rather to evaluate representative members of these families under a *matched* training protocol and optical forward model. We explicitly describe which variants are implemented and how they are trained, to avoid ambiguity in the comparison.

## E.1 GERCHBERG–SAXTON (GS) BASELINE

We first consider the classical GS phase retrieval algorithm as a non-learned baseline. In our setting, GS is run between the SLM plane and the far field, alternately enforcing the measured far-field amplitude and a Gaussian-amplitude/SLM-support constraint in the near field. We tune the number of iterations and step parameters on a validation set, and then fix them for all test cases.

To probe outer-ring generalization, we use the same train/test split as in the main experiments: only 1–3-ring targets are used to choose GS hyperparameters, while evaluation is performed on 4–9-ring test patterns that were never seen during tuning. As illustrated in Figure 2, GS is able to recover a few strong inner rings but systematically fails to reconstruct the correct number and radii of outer rings; instead, it tends to collapse multi-ring structure into a small number of broad peaks. In particular, we did not observe any configuration in which vanilla GS reliably recovered the outermost rings in the presence of distribution shift. We therefore treat GS as a useful qualitative reference, but it does not provide a competitive baseline for the compositional outer-ring generalization task we study.



Figure 2: Example GS reconstruction in the outer-ring generalization setting. GS can recover some inner structure but fails to reproduce the correct number and locations of outer rings when tested on more complex patterns than those seen during tuning.

## E.2 PNP/RED-STYLE LEARNED UNROLLED BASELINES

We next evaluate learned unrolled baselines inspired by prDeep and related PnP/RED methods.

**Architecture and relation to prDeep.** Our PnP/RED baselines are implemented in as the module `PnPREDPhaseNet`. This network follows a prDeep-style unrolled architecture:

- The data-fidelity term uses the same Fraunhofer forward model as in the main paper. Concretely, we define a differentiable operator $A$ (class `FraunhoferPropagator`) that maps a near-field complex field $\psi \in \mathbb{C}^{H \times W}$ to its far-field complex field $A\psi$ via a centered orthonormal FFT. The data term is
$$f(\psi; y) = \tfrac{1}{2} \left\| |A\psi| - y_{\mathrm{amp}} \right\|_2^2,$$
where $y_{\mathrm{amp}}$ is the measured amplitude. The corresponding gradient $\nabla_\psi f$ is implemented in `FraunhoferPropagator.grad_amplitude_loss`.

- The prior term is implemented by a complex DnCNN-style denoiser `ComplexDnCNN` with 17 convolutional layers and 64 feature channels, closely mirroring the DnCNN backbone used in prDeep (Metzler et al., 2018). The denoiser operates on 2-channel real-valued tensors representing the real and imaginary parts of $\psi$.

- We unroll $K$ iterations of a gradient/denoising scheme.

Within this framework we consider two variants:

1. **PnP variant ("PnP" in Table 13):** at iteration $k$ we perform
$$\psi_{k+\frac{1}{2}} = \psi_k - \tau_k \nabla_\psi f(\psi_k; y), \qquad \psi_{k+1} = D(\psi_{k+\frac{1}{2}}),$$
where $D$ is `ComplexDnCNN` and $\tau_k > 0$ is a learnable step size (parameterized via a softplus of `raw_steps[k]`).

2. **RED variant ("RED" in Table 13):** using the same data step, but with a RED-type update
$$\psi_{k+1} = \psi_{k+\frac{1}{2}} - \lambda_k \left(\psi_{k+\frac{1}{2}} - D(\psi_{k+\frac{1}{2}})\right),$$
with $\lambda_k > 0$ learned via softplus of `raw_lams[k]`.

Both variants use the same Fraunhofer operator and denoiser architecture; they differ only in the update rule. We denote these two baselines as *prDeep-style PnP* and *prDeep-style RED* to emphasize their connection to Metzler et al. (2018), while also making clear that they are implemented as learnable unrolled networks.

**Training protocol.** Unlike prDeep, which is typically run as a per-instance solver with a pre-trained denoiser, we train `PnPREDPhaseNet` *end-to-end* on the same synthetic Fraunhofer dataset and outer-ring generalization split as our main Hybrid model. Specifically:

- For each training example, we take the central crop of the far-field complex field $u_{\mathrm{crop}}$ as input and form the amplitude measurement $y_{\mathrm{amp}} = |u_{\mathrm{crop}}|$. The initial guess $\psi_0$ is set to zero; when its magnitude is numerically negligible, we reinitialize $\psi_0$ with $y_{\mathrm{amp}}$ and a random phase.

- The PnP/RED core `PnPREDPhaseNet` is wrapped by `PnPREDWrapper`, which adapts its interface to match the Hybrid model: the output $\psi_K$ is interpreted as a near-field SLM-plane field, combined with the same Gaussian amplitude envelope and parabolic phase, and propagated to the far field using the same `prop_TF_IR` operator.

- We then train the entire unrolled network using the *same* loss and optimization pipeline as the main model: a far-field intensity MSE data term (weight `data_w = 1.0`), optional near-field amplitude and energy terms (weights `amp_w, en_w`), Adam optimizer with learning rate $8 \times 10^{-5}$, cosine schedule, and 60 epochs.

Thus, the PnP/RED baselines share (i) the same optical forward model, (ii) the same training data and outer-ring split, and (iii) a matched training budget with the Hybrid model. This makes the comparison *fair within our setting*, while we do not claim to exactly reproduce all implementation details or performance of Metzler et al. (2018) or diffusion-based RED methods such as Mardani et al. (2023) on their original benchmarks. In particular, we do *not* implement diffusion-model priors; Mardani et al. is cited as a representative example of modern RED formulations, not as an exact baseline in our experiments.

**Quantitative results.** Table 13 summarizes the performance of the prDeep-style PnP and RED baselines in the outer-ring generalization regime. Both models are trained under the protocol above and evaluated using the same global and ring-aware metrics as in Section 6.

Table 13: Performance of prDeep-style PnP and RED baselines (`PnPREDPhaseNet`) in the outer-ring generalization setting. Lower is better for RMSE/EMD/MAE/Outer error; higher is better for SSIM/Precision/Recall/F1/Recovered.

| Metric | PnP | RED |
|---|---|---|
| Rings Profile RMSE ↓ | 0.0268 | **0.0240** |
| Rings Profile EMD ↓ | 0.1362 | **0.1230** |
| Global Intensity RMSE ↓ | 5.4978 | **1.1700** |
| SSIM ↑ | **0.6314** | 0.5887 |
| Phase MAE ↓ | **1.5714** | 1.5716 |
| Ring Precision ↑ | 0.0010 | **0.0087** |
| Ring Recall ↑ | 0.0003 | **0.0028** |
| Ring F1 ↑ | 0.0004 | **0.0039** |
| Recovered Ring Count ↑ | 0.0010 | **0.0140** |
| Hallucinated Ring Count ↓ | **0.0200** | 0.0760 |
| Peak Radius Error ↓ | 0.9870 | **0.9587** |
| GT Ring Count | 6.424 | 6.424 |
| Predicted Ring Count | 0.0210 | 0.0900 |
| Num Samples | 1000 | 1000 |

We observe that, despite having access to the same physics operator and being trained end-to-end, both prDeep-style PnP and RED variants struggle to recover the discrete ring structure in our outer-ring generalization task: ring Precision/Recall/F1 and recovered ring counts are close to zero, and the predicted ring counts are far below the ground truth. At the same time, the global metrics (Rings Profile RMSE/EMD, SSIM, Phase MAE) indicate that the models learn a coarse approximation of the radial energy distribution but fail to correctly instantiate the high-frequency multi-ring pattern, especially for unseen outer rings.

These results suggest that, within the family of CNN-based PnP/RED architectures considered here, simply unrolling a prDeep-style solver and training it end-to-end is not sufficient to obtain strong outer-ring generalization, even under a matched training protocol. This motivates the more specialized Hybrid architecture proposed in the main paper, which explicitly encodes the ring structure and optical PDEs.

# F QUANTITATIVE ROBUSTNESS TO ELLIPTICAL AND ASTIGMATIC ASYMMETRY

To probe how a *strictly radial* architecture behaves when inputs deviate from axisymmetry, we keep the network and training protocol fixed and evaluate it on two controlled families of asymmetric three-ring targets. The model is trained only on perfectly axisymmetric patterns, and its output head enforces radial symmetry. At test time, we introduce an asymmetry parameter $\alpha \in [0, 1]$ that progressively deforms the *target* pattern: (i) elliptical targets use anisotropic horizontal–vertical scaling; and (ii) astigmatic targets use unequal scaling in a $45°$ rotated basis. For each $\alpha$ we propagate the predicted phase through the same diffraction operator and compute near-/far-field MSE, an approximate SSIM, two energy metrics, and two symmetry indicators (radial deviation and Fourier-domain asymmetry).

**Findings.** Across the full range of $\alpha$, near- and far-field MSE remain bounded and the composite performance index degrades smoothly without abrupt failure. Near-field MSE varies only mildly and even decreases at large $\alpha$, consistent with the model output approaching the azimuthal average of increasingly distorted targets; far-field MSE grows with $\alpha$ but saturates. The radial-deviation metric stays essentially zero for all $\alpha$, confirming that the output remains strictly axisymmetric by construction, while the Fourier-domain asymmetry metric increases in line with the imposed ground-truth deformation. Taken together, these results show that the model behaves like a stable projection onto a circular multi-ring manifold: the radial symmetry constraint does not "break down" under

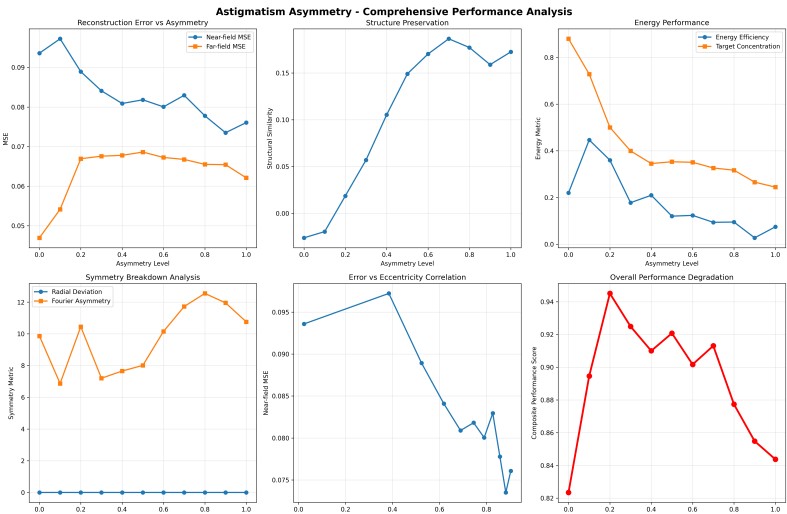

Figure 3: Astigmatism robustness: comprehensive quantitative analysis as a function of asymmetry level $\alpha$.

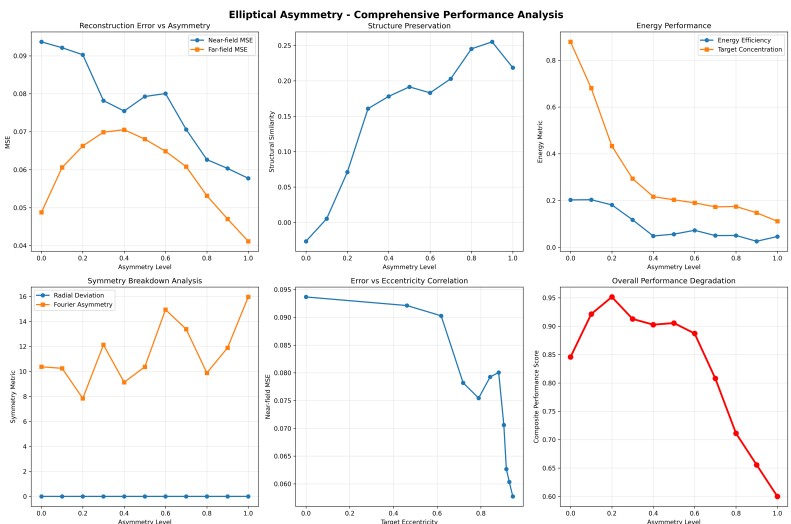

Figure 4: Elliptical robustness: comprehensive quantitative analysis as a function of asymmetry level $\alpha$.

elliptical or astigmatic perturbations, but instead converts asymmetry into a gradual, well-controlled approximation error.

## G  LLM USAGE.

A large language model assistant was used for grammar/style editing, title/caption sentence-case normalization, and LaTeX hygiene (e.g., package conflicts, figure/table formatting). It was *not* used to design experiments, create algorithms, derive theorems, write code that was executed without review, or generate results. All technical content (models, proofs, experiments, analysis) was authored and verified by the authors.

