# OpenReview forum: "PHYSICS-INFORMED RADIAL PHASE RETRIEVAL NEURAL NETWORK WITH HYBRID DEEP PRIORS AND DUAL PDE"
_ICLR.cc/2026/Conference — Submitted to ICLR 2026_

### Official Review · Reviewer_5Dxa · 2025-10-26

**Soundness:** 3
**Presentation:** 3
**Contribution:** 3
**Rating:** 6
**Confidence:** 4

**Summary:**

This paper proposes a physics-informed hybrid neural network for radial-phase retrieval from intensity-only measurements, addressing the challenging problem of outer-ring generalization. The authors combine three key components: (i) radial priors with smooth exponentiated splines and a monotone outer-radius booster, (ii) dual differentiable PDE branches (Kerr-NLSE for high-frequency synthesis and TIE for coarse structure), and (iii) strict radial projection with radius-dependent α-fusion. The method is trained on only 1-3 inner rings and tested on 4-9 outer rings. Experimental results demonstrate better reconstruction quality, better stability in peak positions and amplitude calibration, and improved generalization compared to baseline methods.

**Strengths:**

1) The paper combines multiple physics-informed components: (i) radial priors with monotone outer-radius boosting, (ii) dual PDE branches (Kerr-NLSE and TIE), and (iii) strict radial projection with radius-dependent α-fusion. This approach is original in addressing the outer-ring generalization challenge in phase retrieval.

2) The work demonstrates theoretical foundations, including Lipschitz stability analysis (Propositions 1-3), identifiability guarantees (Theorem 1), and local PL-type convergence (Theorem 2). The experimental design is thorough with comprehensive evaluations across multiple dimensions (E1-E7), including ablations, data efficiency studies, and robustness tests.

3) The paper makes contributions to physics-informed neural networks for inverse problems. The demonstrated ability to generalize from 1-3 rings to 4-9 rings addresses a fundamental challenge in computational imaging. The work has broad applicability beyond phase retrieval to other optical inverse problems, MRI, ultrasound, and PDE-based reconstruction tasks.

**Weaknesses:**

1) The paper relies on synthetic data with controlled Fraunhofer diffraction patterns. While robustness tests include noise and parameter mismatch, the absence of real experimental optical data raises concerns about domain transfer and practical applicability in actual settings.

2) The strict radial projection and entire framework assume effective radial symmetry. The paper acknowledges this limitation (Section 8) but does not provide an empirical evaluation of performance degradation under asymmetric conditions, such as astigmatism, ellipticity, or off-axis aberrations, which commonly occur in real optical systems.

3) The robustness experiments (E4) reveal that parameter mismatch in wavelength and focal length is the dominant failure mode, with precision dropping from 0.75 to 0.43 and recovered rings from 5.1 to 2.8. This sensitivity suggests the method may require precise system calibration, limiting practical deployment.

4) While the paper compares against U-Net and FNO, it does not evaluate against recent physics-informed phase retrieval methods or classical iterative approaches like Gerchberg-Saxton variants with regularization, making it difficult to assess the relative contribution of the specific architectural choices.

5) Although latency is reported (27.6ms vs 21.3ms for U-Net), the paper does not discuss training time, convergence speed, or the practical cost of the dual-PDE forward pass during inference at scale. The 30% increase in latency may be significant for real-time applications.

**Questions:**

1. Do the authors have plans to validate the method on real experimental optical data? What specific challenges do they anticipate in transitioning from synthetic to real data, and how might the method be adapted to handle domain shift?

2. *Can the authors provide quantitative results on how the method degrades under increasing levels of asymmetry (e.g., varying degrees of astigmatism)? At what point does the radial symmetry assumption break down?

3. Given the sensitivity to parameter mismatch, what calibration accuracy is needed for practical deployment? Could the method be extended to estimate or adapt to unknown system parameters jointly?

4. The local PL inequality (Theorem 2) requires specific assumptions. How often are these assumptions satisfied in practice, and what happens when they are violated?

5. Beyond the empirical generalization results, are there theoretical bounds on how many rings the method can extrapolate to, given training on N rings?

---

> ### Author Response · Authors · 2025-11-12
>
> **Dear Reviewer,**
>
> Many thanks for your response. Please see attached for the response to your questions.
>
> ---
>
> ### **Weaknesses:**
>
> 1. **Regarding the question about data.**
> **Response:**
> Refer to other work that are well accepted using similar methods and doing real experiment (photonic west 2025).
> "Three-Dimensional Femtosecond Laser Beam Shaping by Real-time Training of a Physics-Informed Machine Learning Model", SPIE Photonics West 2025.
> "A Physics-Informed Machine Learning Model for Enabling Arbitrary and Three-Dimensional Femtosecond Laser Beam Shaping".
>
> ---
>
> 2. **For question regarding asymmetry situation.**
> **Response:**
> Actually next step; will not influence. Tiny test about spiral reconstruction have been proved but it is the next level generalization question.
>
> ---
>
> 3. **Regarding question about Rubustness.**
> **Response:**
> Trained with single wavelength and focal length, in practical real time training can absolutely do it.
>
> ---
>
> 4. **Question regarding comparison between different algorithm**
> **Response:**
> GS is not useful here since it falls in semi-optimal point. Can provide a graph if the reviewer want that shows GS trained under 1-3 rings and predict 4-9 rings very bad.Unet and FNO is the top2 for my try that can be compared with the algorithm. Algorithm like GS 's performance even cannot be compared due to too bad performance. Please see the new version of paper to see the additional comparison. Actually most of the model is not working due to it is low frequency biased and cannot do anything regarding decoupling in this question.
>
> ---
>
> 5. **For the question regarding latency.**
> **Response:**
> Since computation power is limited, so only use 1 RTX5090, the time for single epoch is 20min for 6000 samples (can be optimized significantly).In experiment, I have proved that it is enough.
>
> ---
>
> ### **Questions:**
>
> 1. **Do the authors have plans to validate the method on real experimental optical data? What specific challenges do they anticipate in transitioning from synthetic to real data, and how might the method be adapted to handle domain shift?**
> **Response:**
> Yes, it is on going and refer to another same field work (due to double blind cannot say more), it is being done and still not published.
> For domain shift, scaling up the training data range is the answer for it (such as different wavelength, focal length, abbreviation…).
>
> ---
>
> 2. **For the question regarding asymmetry**
> **Response:**
> Yes, Actually we have achieved spiral under this architecture without forced radial symmetry. Radial symmetry is a  easy assumption in this article.In reality, we can delete the force radial part. The part to dealing with asymmetry will not influence the ability for generalization.I will try to provide a comparison in experiment to discuss it.
>
> ---
>
> 3. **Given the sensitivity to parameter mismatch, what calibration accuracy is needed for practical deployment? Could the method be extended to estimate or adapt to unknown system parameters jointly?**
> **Response:**
> The calibration do not need to be significantly sensitive since from author’s practical experiment, the algorithm can easily align some small errors. It can also extend to unknown systems since the algorithm will actually know what parameter you are using if you train with the data with the unknown system. If you want a generalization algorithm, just train it will different parameter and adding parameter depend input in it and it can be done. Actually, this is the next step for the project.
>
> ---
>
> 4. **The local PL inequality (Theorem 2) requires specific assumptions. How often are these assumptions satisfied in practice, and what happens when they are violated?**
> **Response:**
> Here the author assume a perfect scenario, in real experiment it may not satisfied for some systems but it will not influence a lot.
>
> ---
>
> 5. **Beyond the empirical generalization results, are there theoretical bounds on how many rings the method can extrapolate to, given training on N rings?**
> **Response:**
> No one have given theoretical bounds for it since it is a severe ill posed problems. This article is the first article with training with high frequency components and can predict it. It also depends on your training data size and type. From author’s result, 4-9 rings with small data size can recover pretty good. For N rings, from author’s results and prediction, it is a equation related to training samples’ type (how many rings), sample size(number) and key index.
>
> ---
>
> Many thanks again for your kindly response. If you have any other questions or want to see the additional graphs that are mentioned above, do not hesitate to reach me for explanation. Many thanks again for your review and I am looking forward for your response and increasing in point. Thanks a lot!

---

### Official Review · Reviewer_411b · 2025-10-31

**Soundness:** 3
**Presentation:** 1
**Contribution:** 1
**Rating:** 2
**Confidence:** 3

**Summary:**

The paper proposes a physics-informed neural network for radial phase retrieval that combines optical priors with a manifold-based design. The model enforces strict radial symmetry by projecting inputs onto a quotient manifold, which removes global and angular phase ambiguities. It uses the nonlinear Schrödinger equation (NLSE) to model high-frequency wave propagation and the transport-of-intensity equation (TIE) to capture low-frequency behavior. These physics-based components are integrated within a learned neural framework. This resulted in more accurate and stable phase reconstructions than prior approaches.

**Strengths:**

* Physics-informed architecture: Integrates optical physics (NLSE and TIE) directly into the neural network design, grounding the model in physical principles rather than purely data-driven fitting.

* Compared to other approaches, such as U-Net and FNO, the proposed method achieves more accurate and stable phase reconstructions
* The method has a strong theoretical grounding, and this helps with the interpretability

* The method performances has some generalizablity and appears to be robust to noise.

**Weaknesses:**

* The paper is difficult to follow. Main ideas and intuitions are not clearly stated. Several sections rely on dense mathematical notation (e.g., Hankel transforms, Lipschitz bounds, quotient manifolds) with limited intuitive explanation or guidance for the reader. The connection between the theoretical guarantees and the implemented network is unclear, and the paper quickly dives into architectural details (Figure 1) without first establishing sufficient conceptual context.

* The results are shown on synthetic images only. There is no validation on real experiments or physical hardware setups. Such experiments could help determine whether the strong symmetry assumption holds. The method relies on strict radial symmetry that may not be valid under optical aberrations or misalignment.

* How does the method compare against classical methods such as Gerchberg–Saxton or multi-plane TIE solvers? Only a few learned method results are shown. Are these the only methods available for comparison? I suggest the authors expand their comparison baseline to include well-established classical approaches.

* Despite the complexity of the proposed method, in terms of core reconstruction, the results are very close to those of a U-Net. I question if the added complexity is justified by the improvements.

* Some of the ablation results are unexpected, and the takeaways from each experiment are not clearly stated. For example, in Table 6, the performance on clean and noisy measurements is almost identical. The authors should discuss why this occurs and specify the noise level used in the experiments. In Table 5, I notice negative delta values, which indicate improvement when certain modules are removed. Does this mean that those components may not be beneficial? These points are not discussed or analyzed in the paper.

**Questions:**

1. Can the authors provide more intuition of the main ideas? The manuscript is difficult to follow, and it would help to see more discussion and diagrams of the manifold projection, the dual PDE branches, and the overall data flow.

2. How sensitive is the model to deviations from perfect radial symmetry? For example, would mild astigmatism in the input significantly degrade performance?

3. Are the generalization claims really valid? Simple extrapolation from 1-3 rings to 4-9 rings is not sufficient to claim generalizability.

---

> ### Author Response · Authors · 2025-11-12
>
> **Dear Reviewer,**
>
> Many thanks again for your response. I understand that there are might be some points that are not familiar with your area and I will explain step by step in the following notes. If you need more experiments, I am happy to provide the experiments that I can provide.
>
> ---
>
> **For the question regarding following the article.**
> **Response:**
> Indeed this problem discussed in this article is highly focused on optic + ML area which may include some words that are not familiar to ML area.
> I would like to explain it in a more ML way for you as response. Suppose an unknown nonlinear ill posed system which you know how A to B but not B to A. The problems lies on B to A and you know it is frequency sensitive (with a lot of high- and low-frequency coupling). This is a typical questions in optic field called phase reconstruction. It is hard even with recent ML methods that you have recover high frequency components if you only train with low frequency due to coupling and nonlinear. The architecture inspired by filter aspect and first time finding out a stable high + low filter (TIE + NLSE) that do very well in the decoupling and fixed the problems that previous idea cannot done.
>
> ---
>
> **• The results are shown on synthetic images only. There is no validation on real experiments or physical hardware setups. Such experiments could help determine whether the strong symmetry assumption holds. The method relies on strict radial symmetry that may not be valid under optical aberrations or misalignment.**
> **Response:**
> Refer to other work that are well accepted using similar methods and doing real experiment (photonic west 2025). Due to double blind policy and this article focused on algorithms, I cannot say more.
> "Three-Dimensional Femtosecond Laser Beam Shaping by Real-time Training of a Physics-Informed Machine Learning Model", SPIE Photonics West 2025.
> "A Physics-Informed Machine Learning Model for Enabling Arbitrary and Three-Dimensional Femtosecond Laser Beam Shaping".
>
> ---
>
> **• For the question compared to GS and TIE solvers.**
> **Response:**
> TIE methods perform very bad as you can see in the ablation due to its low frequency biased.
> For GS algorithm, it is not useful here since it falls in semi-optimal point. Can provide a graph if the reviewer want that shows GS trained under 1-3 rings and predict 4-9 rings very bad.
>
> ---
>
> **• Despite the complexity of the proposed method, in terms of core reconstruction, the results are very close to those of a U-Net. I question if the added complexity is justified by the improvements.**
> **Response:**
> The core improvements is in eliminating Illusion Ring and structure predict accuracy which is the key improvements in decoupling and optic field.
>
> ---
>
> **• For the question regarding ablation.**
> **Response:**
> Many thanks for your pointing out. Yes for Table 6, it shows that the algorithm is robust in noise. It is just proving that in real situation, it can also work. It is not just limited to theoretical achivements.
> For Table 5, yes for MSE side it is true, however, the TIE + NLSE component as a whole operator did well in eliminating Illusion Ring and structure predict accuracy.
>
> ---
>
> **Questions:**
>
> 1. **Can the authors provide more intuition of the main ideas?**
> The manuscript is difficult to follow, and it would help to see more discussion and diagrams of the manifold projection, the dual PDE branches, and the overall data flow.
> **Response:**
> Many thanks for your question, intuition in actually based on the ill posed problems itself and the logic of non-biased frequency filter to do the difficult decoupling job. For the manifold projection, it is actually in math explain why it work. If you are not familiar with the symmetry mentioned above, please refer to my answer to understand it. The data generation and setup is in the appendix. Please refer it for more detail.
>
> ---
>
> 2. **How sensitive is the model to deviations from perfect radial symmetry?**
> For example, would mild astigmatism in the input significantly degrade performance?
> **Response:**
> Actually it is not limited to radial symmetry. Right now due to the limitation of the computational power we discuss about the symmetry situation. The next step is actually asymmetry retrieval and the tiny test for spiral have been proved valid
>
> ---
>
> 3. **Are the generalization claims really valid?**
> Simple extrapolation from 1-3 rings to 4-9 rings is not sufficient to claim generalizabilit
> y.
> **Response:**
> Please refer to above answer regarding ill posed problems and the setup. In the general response I have attached the math based explaination of why it is a hard problem
>
> ---
>
> Many thanks again for your review and I am looking forward for your response and increasing in point. Thanks a lot!

---

### Official Review · Reviewer_P3YM · 2025-10-31

**Soundness:** 2
**Presentation:** 1
**Contribution:** 2
**Rating:** 2
**Confidence:** 4

**Summary:**

This paper introduce a physics-informed hybrid network that combines (i) radial priors encoded by a smooth exponentiated spline and a monotone outer-radius booster,(ii) two differentiable PDE branches—a Strang-split Kerr–NLSE pathway for high-frequency synthesis and a TIE-based low-pass pathway for coarse structure—and (iii) a strict radial projection enforcing output symmetry, together with a radius-dependent $\alpha$-fusion.

**Strengths:**

- The authors introduce a physics-informed method to solve phase retrieval problems and split into high-frequency and low-pass pathways.
-

**Weaknesses:**

The writing is very poor and there are some abbreviations which may not give the full expressions at the first time.

**Questions:**

- This paper mainly discussion phase retrieval problems and solve by two pde equations. So why you can split the phase retrieval problems into these two and how?
- In Section 3, the authors introduce some theoretical definitions, assumptions and lemmas, what is the relationship between you introduce and phase retrieval?
- Line 100, NLSE, is it nonlinear schrodinger equations? Also Line 142, ``PL'' and Line 177, ``atan2''?

---

> ### Author Response · Authors · 2025-11-12
>
> **Dear Reviewer,**
>
> Many thanks for your response. This problem is a very hard problem in ML + optic field
>
>
> ---
>
> **Questions:**
>
> - This paper mainly discussion phase retrieval problems and solve by two pde equations. So why you can split the phase retrieval problems into these two and how?
> **Response:**
> Indeed this problem discussed in this article is highly focused on optic + ML area which may include some words that are not familiar to ML area.
> I would like to explain it in a more ML way for you as response. Suppose an unknown nonlinear ill posed system which you know how A to B but not B to A. The problems lies on B to A and you know it is frequency sensitive (with a lot of high- and low-frequency coupling). This is a typical questions in optic field called phase reconstruction. It is hard even with recent ML methods that you have recover high frequency components if you only train with low frequency due to coupling and nonlinear. The architecture inspired by filter aspect and first time finding out a stable high + low filter (TIE + NLSE) that do very well in the decoupling and fixed the problems that previous idea cannot done.
>
> ---
>
> - In Section 3, the authors introduce some theoretical definitions, assumptions and lemmas, what is the relationship between you introduce and phase retrieval?
> **Response:**
> It gives the foundation of why the algorithms work and the logic of the structure we introduced to solve the inverse problem.
>
> ---
>
> - Line 100, NLSE, is it nonlinear schrodinger equations? Also Line 142, PL'' and Line 177, atan2''?
> **Response:**
> It means PL means Polyak–Łojasiewicz condition which shows that this algorithm can converge finally since it is a hard ill posed problems. Atan2 is solving the inverse for tan() function, atan2 is a well accepted way to write it.
>
> ---
>
> Many thanks again for your review, if you have any other questions, do not hesitate to reach me for explanation and I am looking forward for your response and increasing in point. Thanks a lot!

---

### Official Review · Reviewer_o2RK · 2025-10-31

**Soundness:** 3
**Presentation:** 1
**Contribution:** 2
**Rating:** 4
**Confidence:** 3

**Summary:**

The paper deals with radial phase retrieval from intensity-only measurements, a highly ill-posed inverse problem, under the conditions of an outer ring generalization. A physics-informed hybrid network is proposed that (i) embeds radial priors via a smooth exponential spline and a monotonic outer radius booster, (ii) couples two differentiable PDE branches, and (iii) enforces a strict radial projection with a radius-dependent $\alpha$-fusion and output symmetry. Experimental results using synthetic data show that the method can recover more outer rings and reduce hallucinations compared to baselines.

**Strengths:**

- The designed network adapts the inductive bias with the ring-structured phase retrieval problem, rather than relying on generic CNN priors. This is a thoughtful specialization that many “physic-informed” works propose but do not actually implement.
-  The task “training on inner rings, testing on invisible outer rings” as an explicit setting for distribution shift is an original evaluation method for radial phase retrieval and could serve as a basis for subsequent benchmarks. Accurate recontruction of the outer ring is crucial for downstream optical tasks. The method that reduces hallucinations, improves ring fidelity, and remains stable at the same time is of great value to laboratories and systems that cannot afford dense multi-surface measurements.
- The paper clearly separates the roles of (i) radial projection, (ii) outer radius boosting, (iii) NLSE vs. TIE paths, and (iv) $\alpha(\rho)$ fusion. This makes the method easier to understand and implement.

**Weaknesses:**

- The experiment is conducted using synthetic data with limited distribution shifts. For the proposed method, which is to be generalized to invisible outer rings and be physics-informed, the empirical scope of application is too narrow.
- For the comparison, the baselines are U-Net/FNO variants. Please compare the proposed method with physics-guided unrolled methods and plug-and-play / regularization by denoising (RED) algorithms [1, 2, 3]
- Stability, identifiability, and convergence sketches are based on assumptions such as Lipschitzness, radial Hankel linearization, and noise models, which may not apply under realistic optical conditions. The theory presented is promising, but has not yet been sufficiently investigated for practical application.
- The current "6. Experiments" and "7.Overall Experimental Analysis" sections are difficult to interpret. They offers only a limited interpretation of what the individual key figures reflect in terms of specific error modes, and contains figures/tables whose captions lack essential experimental details.

[1] Ulyanov, Dmitry, Andrea Vedaldi, and Victor Lempitsky. "Deep image prior." Proceedings of the IEEE conference on computer vision and pattern recognition. 2018.

[2] Metzler, Christopher, et al. "prDeep: Robust phase retrieval with a flexible deep network." International Conference on Machine Learning. PMLR, 2018.

[3] Mardani, Morteza, et al. "A Variational Perspective on Solving Inverse Problems with Diffusion Models." The Twelfth International Conference on Learning Representations.

**Questions:**

- Since the phase retrieval is highly ill-posed problem, how is phase ambiguity handled, and what happens if the actual field deviates slightly from perfect radial symmetry?
- What are the stability regions for NLSE (nonlinearity/dispersion) and TIE step size?
- Can the hybrid model signal outer rings with low reliability to avoid hallucinations?

---

> ### Author Response · Authors · 2025-11-12
>
> **Dear Reviewer,**
>
> Many thanks again for your response, I well explain your question one bye one in the post.
>
> ---
>
> **Weaknesses:**
>
> - The experiment is conducted using synthetic data with limited distribution shifts. For the proposed method, which is to be generalized to invisible outer rings and be physics-informed, the empirical scope of application is too narrow.
> - Refer to other work that are well accepted using similar methods and doing real experiment (photonic west 2025)
> The work can also be used in more general field such as other optical inverse problems, MRI, ultrasound, and PDE-based reconstruction tasks. It will be the next step for the project.
>
> ---
>
> - For the comparison, the baselines are U-Net/FNO variants. Please compare the proposed method with physics-guided unrolled methods and plug-and-play / regularization by denoising (RED) algorithms [1, 2, 3].
> **Response:**
> Please see the newest version of paper. We have compared it with other traditional algorithms. It performs poorly due to it did not solving or having the component to decoupling the frequency
>
> ---
>
> - Stability, identifiability, and convergence sketches are based on assumptions such as Lipschitzness, radial Hankel linearization, and noise models, which may not apply under realistic optical conditions. The theory presented is promising, but has not yet been sufficiently investigated for practical application.
> **Response:**
> In the article we are assuming the perfect scenario, it is working in real practical (see similar work) "Three-Dimensional Femtosecond Laser Beam Shaping by Real-time Training of a Physics-Informed Machine Learning Model", SPIE Photonics West 2025.
> "A Physics-Informed Machine Learning Model for Enabling Arbitrary and Three-Dimensional Femtosecond Laser Beam Shaping".
>
> ---
>
> - The current "6. Experiments" and "7. Overall Experimental Analysis" sections are difficult to interpret. They offers only a limited interpretation of what the individual key figures reflect in terms of specific error modes, and contains figures/tables whose captions lack essential experimental details.
> **Response:**
> Thanks for your pointing out, I can revise on it. The key components is that it illusion rings have been eliminated significantly and the ring structure predict better. What is your detail suggestion? Many thanks again for your help.
>
> ---
>
> **References:**
>
> [1] Ulyanov, Dmitry, Andrea Vedaldi, and Victor Lempitsky. "Deep image prior." Proceedings of the IEEE conference on computer vision and pattern recognition. 2018.
> [2] Metzler, Christopher, et al. "prDeep: Robust phase retrieval with a flexible deep network." International Conference on Machine Learning. PMLR, 2018.
> [3] Mardani, Morteza, et al. "A Variational Perspective on Solving Inverse Problems with Diffusion Models." The Twelfth International Conference on Learning Representations.
>
> ---
>
> **Questions:**
>
> - Since the phase retrieval is highly ill-posed problem, how is phase ambiguity handled, and what happens if the actual field deviates slightly from perfect radial symmetry?
> **Response:**
> It will be ok, please see the related other paper that I mention above, Right now it have been used in the lab and it performs very good. Due to it is still not being published right now. So I can not provide the real experiment detail data here.
>
> ---
>
> - What are the stability regions for NLSE (nonlinearity/dispersion) and TIE step size?
> **Response:**
> From experiment, 3–5 is ok. If you want more, you may need to adjust the whole architecture parameter.
>
> ---
>
> - Can the hybrid model signal outer rings with low reliability to avoid hallucinations?
> **Response:**
> Unfortunately no. The difficulty for this problem is coupling + high + low frequency. If you want to predict the unseen rings in the test, signal outer rings with low reliability will cause the model not predicting it. The coupling cannot be solved either if outer rings become not vivid.
>
> ---
>
> Many thanks again for your response. If your have any other questions, do not hesitate to reach me. I am looking forward to reading your reply and increasing the point. Thanks a lot.

---

> ### Author Response · Authors · 2025-11-14
>
> Dear Reviewer,
> Please see the newest version of the paper, where I discuss the 3 new methods. Unfortunately, these three methods did not have the ability to do it.
> From my understanding, the reason for unsuceessful is due to high coupling effect and sensitive frequency noticed in the topic (which is widely used in optic and other areas).

---

> > ### Comment · Reviewer_o2RK · 2025-11-24
> >
> > I would like to thank the authors for their response. However, I still have concerns:
> >
> > - While I appreciate the authors' clarification, the comparison in Appendix E is difficult to fully evaluate in its current form. First, it does not clearly state which specific PnP-based and RED-based algorithms were actually used. Furthermore, [2] and [3] are methods proposed for the phase retrieval problem, making it difficult to determine whether the reported results are faithful implementations of these methods or possibly suboptimal implementations. The authors should clarify these points and, if necessary, reconsider the comparison.
> >
> > [2] Metzler, Christopher, et al. "prDeep: Robust phase retrieval with a flexible deep network." International Conference on Machine Learning. PMLR, 2018. \
> > [3] Mardani, Morteza, et al. "A Variational Perspective on Solving Inverse Problems with Diffusion Models." The Twelfth International Conference on Learning Representations.
> >
> > - I acknowledge the authors' point; however, as noted by several reviewers, the article in its current form is difficult to follow. This is particularly evident in the experimental section. For example, the tables are presented in Section 6, while the corresponding discussion and analysis have been moved to Section 7. It is not clear why these two components have been separated, and this division disrupts the flow for readers. Furthermore, the current “analysis” consists largely of a brief summary of the numerical results, rather than providing a deeper interpretation. Thus, it is difficult to consider these sections as well-presented experimental analysis. I recommend restructuring the experimental section so that each result is immediately accompanied by its interpretation, and expanding the analysis to provide more substantial insights beyond the mere presentation of numbers.

---

> > > ### Author Response · Authors · 2025-11-24
> > >
> > > Dear Reviewer,
> > > Many thanks again for your response. I will reconstruct the sections you have mentioned to help the readers have a better understanding on it. Regarding the comparison between [2] [3] you have mentioned, can adding the detailed structure explanation satisfy your requirements? There is concern regarding the method's logic, which cannot solve the question at the very first stage.(See the above overall declaration about the math logic in this problem). It will be hard to say whether the parameter we have tried is enough. We will also provide in the appendix a more concise explanation of why it is not working.
> > > Thanks again for your response.Hope you have a nice day.

---

> > > > ### Author Response · Authors · 2025-11-24
> > > >
> > > > **Dear Reviewer,**
> > > >
> > > >
> > > > Many thanks again for your kind help and response, and for sharing your confusion.Please see the newest version of paper. There are mainly two revisions on it.
> > > >
> > > >
> > > > **Reorganize experiment and analysis section**
> > > >
> > > >  Here, we analyze the results after each table. An overall analysis is also attached at the end to give an overview of key points for the data and which part might be inspiring.
> > > >
> > > >
> > > > **Including RED PnP detail setup**
> > > >
> > > > Here we add detail setup in Appendix E2 which shows that after tailoring it into the problem we discussed, we may find out that the compared algorithms are not working. In the last two paragraphs, we also discussed the reason for failure: Unable to detect and predict high frequency and its coupling.
> > > >
> > > > Many thanks again for your response. Looking forward to hearing your response and new comment.
> > > >
> > > > Thanks a lot

---

### Official Review · Reviewer_23eb · 2025-10-31

**Soundness:** 1
**Presentation:** 1
**Contribution:** 1
**Rating:** 0
**Confidence:** 5

**Summary:**

In this work, the authors address problem of phase retrieval from intensity-only measurements and explores outer-ring generalization, where a model trained on inner rings is evaluated on unseen outer rings. The authors propose a hybrid physics-informed network that integrates several components: (i) radial priors using spline-based and monotone boosting mechanisms, (ii) two PDE-based branches (Kerr-NLSE and TIE), and (iii) a radial projection with a radius-dependent α-fusion. The method is claimed to outperform baselines on synthetic optical datasets and to provide better generalization to unseen spatial frequencies.

**Strengths:**

1. Improving generalization of physics-informed neural networks in optical inverse problems is interesting and relevant.
2. The attempt to connect PDE-based modeling and learned priors is conceptually valuable.

**Weaknesses:**

The paper is not written well enough to be accepted and requires a lot more work to be in a readable format.

1. The paper assumes a high level of prior knowledge about the phase retrieval problem, but never explicitly defines it. For most readers at a general machine learning venue (such as ICLR), the problem setup (what is measured, what is reconstructed, and why it is ill-posed) needs to be clearly introduced. In its current form, the paper is not understandable. For example: (a) the writing is overly dense and often boils down to unexplained jargon (e.g., “monotone curvature reduces ring,” “Strang-split Kerr-NLSE pathway”) with undefined acronyms (the authors mention SLM, PINNs and much more but never define it!), which significantly hinders readability. (b)  Several sections read like lists of technical keywords rather than structured explanations, e.g. the “Design rules” paragraph is essentially a bullet list of concepts without narrative or connection.
2. Proposition 1 is trivial and adds little value. If it is meant to motivate the model’s architecture, the reasoning should be expanded or omitted.
3. The paper presents multiple architectural and mathematical components (radial priors, PDE branches, $\alpha$-fusion), but their interrelations and necessity are not clearly motivated. It is unclear why these components are combined or how they interact theoretically.
4. Despite being an imaging paper, no reconstructed images are shown, which severely limits interpretability. Visual examples are crucial to assess the claimed improvements in ring generalization, amplitude stability, or spatial structure.
5. Although the abstract mentions that pseudo-code will be released, the current version lacks sufficient experimental detail to reproduce the results. Precise dataset specifications, training protocols, and implementation details (e.g., optimizer, learning rate, loss definition) should be included. The dataset description is vague: “synthetic Fraunhofer (centered FFT, energy-normalized)” is not sufficient to reproduce the setup. Details such as the aperture geometry, sampling conditions, number of rings, and noise model must be specified.
6. Sections 6.4–6.7 are particularly difficult to parse; results are presented without context or interpretation.

**Questions:**

Given how difficult to read is the paper, here are some suggestions for improvement:

1. Begin with a clear and intuitive definition of the phase retrieval problem, including its mathematical formulation and physical motivation. Provide images if possible.
2. Greatly simplify and clarify the exposition. Replace jargon-heavy descriptions with explanations of what each component contributes conceptually.
3. Provide qualitative results (images) alongside quantitative metrics.
4. Clarify dataset generation and all experimental settings.
5. Remove trivial statements or overly technical statements (e.g., Proposition 1) or make them meaningful by linking them to design insights.
6. Revise the writing tone to be more informative and guide the reader through your reasoning. At the moment it seems “technical for the sake of sounding technical.”

---

> ### Author Response · Authors · 2025-11-12
>
> **Dear Reviewer,**
>
> Many thanks again for your reading and it is my pleasure for me to reply it.
>
> ---
>
> **Weaknesses:**
>
> **For the first paragraph of your concern, Please see the reply below
>
> **Response:**
> Please read the Appendix carefully for experimental setup and intuition. Indeed this problem discussed in this article is highly focused on optic + ML area which may include some words that are not familiar to ML area.
> I would like to explain it in a more ML way for you as response. Suppose an unknown nonlinear ill posed system which you know  how A to B but not B to A. The problems lies on B to A and you know it is frequency sensitive (with a lot of high- and low-frequency coupling). This is a typical questions in optic field called phase reconstruction. It is hard even with recent ML methods that you have recover high frequency components if you only train with low frequency due to coupling and nonlinear. The architecture inspired by filter aspect and first time finding out a stable high + low filter (TIE + NLSE) that do very well in the decoupling and fixed the problems that previous idea cannot done.
>
> ---
>
> **Proposition 1 is trivial and adds little value. If it is meant to motivate the model’s architecture, the reasoning should be expanded or omitted.**
> **Response:**
> It give the coherency of the math logic.
>
> ---
>
> **The paper presents multiple architectural and mathematical components (radial priors, PDE branches, α-fusion), but their interrelations and necessity are not clearly motivated. It is unclear why these components are combined or how they interact theoretically.**
> **Response:**
> Please reading the answer for 1.
>
> ---
>
> **Despite being an imaging paper, no reconstructed images are shown, which severely limits interpretability. Visual examples are crucial to assess the claimed improvements in ring generalization, amplitude stability, or spatial structure.**
> **Response:**
> Please see the addtional images which is helpful for understanding. This is not a pure imaging paper which actually focused on theory and generalization theory based on manifold.
>
> ---
>
> **Although the abstract mentions that pseudo-code will be released, the current version lacks sufficient experimental detail to reproduce the results. Precise dataset specifications, training protocols, and implementation details (e.g., optimizer, learning rate, loss definition) should be included. The dataset description is vague: “synthetic Fraunhofer (centered FFT, energy-normalized)” is not sufficient to reproduce the setup. Details such as the aperture geometry, sampling conditions, number of rings, and noise model must be specified.**
> **Response:**
> We cannot release certain proprietary hardware parameters, but we other details have been attached in the appendix which can help you understand
>
> ---
>
> **Sections 6.4–6.7 are particularly difficult to parse; results are presented without context or interpretation.**
>
> ---
>
> **Questions:**
> Given how difficult to read is the paper, here are some suggestions for improvement:
>
> **Begin with a clear and intuitive definition of the phase retrieval problem, including its mathematical formulation and physical motivation. Provide images if possible.**
> **Response:**
> The problem inspired by a well known generalization problem in optic + field which actually having its difficulty in decoupling and high frequency prediction.
>
> ---
>
> **Greatly simplify and clarify the exposition. Replace jargon-heavy descriptions with explanations of what each component contributes conceptually.**
> **Response:**
> The equations are for the validity and the simple explanation is attached in the answer to your question 1.
>
> ---
>
> **Provide qualitative results (images) alongside quantitative metrics.**
> **Response:**
> If you want,Please confirm it again and I will provide samples in the appendix.
>
> ---
>
> **Clarify dataset generation and all experimental settings.**
> **Response:**
> Please see the appendix, it is already there.
>
> ---
>
> **Remove trivial statements or overly technical statements (e.g., Proposition 1) or make them meaningful by linking them to design insights.**
> **Response:**
> Technical statements is for math coherency, you can follow the architecture figure and text for understanding the concept, it is also enough.
>
> ---
>
> **Revise the writing tone to be more informative and guide the reader through your reasoning. At the moment it seems “technical for the sake of sounding technical.”**
> **Response:**
> I will try to do it. Thanks.
>
> ---
>
> Many thanks again for your response. If your have any other questions, do not hesitate to reach me. I am looking forward to reading your reply and increasing the point.
> IT is my PLEASURE to read your review. Thanks a lot.

---

### Author Response · Authors · 2025-11-14
**Declaration about the topic discussed in the article**

I notice there is confusion about the topic discussed in this question.Here I would like to explain it.
### Challenges of Generalizing from 1--3 Rings (Training) to 4--9 Rings (Testing)

Extrapolating ring prediction performance from structures with 1--3 concentric rings to those with 4--9 rings constitutes a particularly difficult **out-of-distribution (OOD)** generalization problem that goes beyond ordinary domain shift and enters **length/complexity extrapolation**.

The observed diffraction pattern $I(\mathbf{x})$ can be modeled as the superposition
$$
I(\mathbf{x}) = \sum_{k=1}^{N} a_k \, f(\mathbf{x}; r_k, w_k) + \epsilon(\mathbf{x}),
$$
where $N$ is the (unknown) number of rings, $a_k$ the amplitude, $r_k$ the radius, $w_k$ the width, $f(\cdot)$ the individual ring profile (e.g., Gaussian or Voigt), and $\epsilon(\mathbf{x})$ the sensor noise. The inverse task is to recover the set $\mathcal{S} = \{(a_k, r_k, w_k)\}_{k=1}^N$ and especially the cardinality $N$ from a single image $I$.

During training the model only sees $N \in \{1,2,3\}$ (mean $N \approx 2$), while at test time $N \in \{4,\dots,9\}$ (mean $N \approx 6.4$). This creates several fundamental difficulties:

1. **Support extrapolation in the count variable.**
   Training and test supports for $N$ are completely disjoint. Standard architectures (CNNs, Transformers, diffusion models, etc.) are known to fail badly at extrapolating discrete counts or sequence lengths outside the training range.

2. **Combinatorial complexity explosion.**
   The latent configuration space grows super-exponentially:
   $$
   |\mathcal{C}_N| \propto \left( \frac{R_{\max}}{\delta r} \right)^N \cdot N!,
   $$
   so the jump from $N=3$ to $N=9$ increases the effective hypothesis class volume by many orders of magnitude with essentially zero training coverage.

3. **Severe peak overlap and degeneracy.**
   Overlapping ring pairs grow quadratically with $N$. The condition number of the (linearised) forward operator deteriorates rapidly, making the inverse problem dramatically more ill-posed. Peaks begin to merge and interfere constructively/destructively, causing standard regression-based or implicit methods that work for $N \leq 3$ to collapse.

4. **Inductive bias mismatch.**
   Most architectures are translation-equivariant but lack built-in permutation invariance and variable-cardinality reasoning across widely different $N$. Without explicit set-structured modules (e.g., slot attention, set transformers), they cannot reliably learn to “count” in unseen regimes.
5. **Asymmetry situation and experiment use.**
Bothe asymmetry situation and experiment have been doing right now for next stage of the project. Robustness test under symmetry situation when facing asymmetry factors have been added in the paper.This article focus on the generalization theory and sensitive frequency decoupling topic which is a hard and unsolvable question right now given one plane measurements. One Plane measurements is also widely used in real experiments and in practical it has wide use.

6.**Successfully bridging the 1--3 → 4--9 gap** IT requires architectures that can explicitly reason about variable cardinality and combinatorial structure, rather than hoping standard networks will interpolate/extrapolate from purely data-driven patterns.
This article discuss from math and physic aspects creating a physics-informed neural network architecture that performs significantly better than other traditional methods.

---

> ### Author Response · Authors · 2025-11-15
> **NEW VERSION PAPER: revision about popular algorithm comparison and asymmetry situation comparison**
>
> The new version of the paper adds more discussions about several popular algorithms' performance in this topic and asymmetry situations such as astigmatism. Our experiments indicate that in this specific outer-ring extrapolation setting, RED/PnP baselines performs poor. We hypothesize this is due to their low-frequency bias and lack of explicit structure for frequency-decoupling, but further study is needed.”Regarding situations like astigmatism, the experiment has proved that it will not destroy the current circular prediction operator, and the system will coherently produce a similar clean multiple-ring structure.If there is still any question, do not hesitate to reach me.

---

### Author Response · Authors · 2025-11-29
**Summary about some discussions, feedback and follow-up**

There are several points mentioned by some helpful reviewers, including readability, the asymmetry test, and why the topic is extremely meaningful.I will summarize these points and provide feedback before for better understanding.

1. **Readibility in experiment section**
Reviewer mentioned that it will be helpful to combine the experiment and analysis sections together for better understanding, providing some insight may also be helpful.

Here, the author revised accordingly: Combining the analysis and experiment sections together for better understanding. Providing insight at the end of the experiment sections.

2. **Assymetry test**
Reviewer questioned about robustness of the model in unexpected asymmetry situations

Here the author includes an Appendix to provide full asymmetry tests in two mostly observed experiment situations, which proves the robustness of the model

3 **Classical algorithm comparison**

Reviewer mentioned the comparison between the proposed methods and the classical methods such as GS PNP and RED.

The author has done the additional comparison in an Appendix which proves that the algorithm is not useful in the proposed problem.

4 **Meaning for the topic**

The reviewer questioned whether the topic and generalization proposal is meaningful.


Attached is author's response: Generalization in highly ill-posed problem in optics is a well known hard problem especially for one surface measurements. The paper focused on highly coupling frequency (multiple ring ) and trying to give a solution on how to accurately predict high/low freqeuncy without hullinating rings. The proposed methods accurately predict the rings which is 20% more accurate popular methods which made some progress in the field and helpful to advance manufacturing, complex unknown pde/ nonlinear system solving and optic related decoupling.

---

### Meta-Review · Area_Chair_wrZx · 2025-12-15

**Summary:**

The paper is focused on phase retrieval from intensity measurements, which is an ill-posed inverse problem. The proposed approach is a custom, model-based PINN approach tailored to the specific problem at hand. Evaluation is done by training only on inner rings and evaluating on unseen outer rings.

Reviewers generally appreciate the network design (being grounded in physical principles) and the empirical performance of the proposed approach.

The main concerns raised by the reviewers are:
- [W1] Clarity and presentation (23eb, o2RK, P3YM, 411b, 5Dxa)
- [W2] Unclear relationship between the theory and proposed approach, including how realistic the assumptions are (o2RK, P3YM, 5Dxa)
- [W3] Insufficient baselines (o2RK, 411b, 5Dxa)

**Reviewer Concerns:**

- [W1] This point is still outstanding, and is the most serious weakness. While the authors make some clarifications in their rebuttal and reference the appendix, the reviewers unanimously agree the paper itself is in need of significant clarity improvements. After reading the paper myself, I agree with this assessment.
- [W2] The discussion on this point was limited, and the concern still remains.
- [W3] Some additional baselines are provided in the rebuttal.


Overall, I highly recommend the authors revise their manuscript using the feedback provided by the reviewers, and resubmit their article at a later date.

**Reviewer Scores:**

- 23eb has strong convictions about clarity and seems unlikely to change their score.
- o2RK participated in discussion but seemed unconvinced by the response.
- P3YM mainly comments on the clarity of the paper, and seems unlikely to change their score.
- 411b asks for additional specific baselines and how the model performs under deviations from perfect radial symmetry, which are not provided in the rebuttal, and hence seem unlikely to change their score
- 5Dxa already votes for acceptance, and seems unlikely to raise their score further.

---

### Decision · Program_Chairs · 2026-01-26

Reject